# Molecular distributions of dicarboxylic acids, oxocarboxylic acids and α-dicarbonyls in PM$_{2.5}$ collected at the top of Mount Tai, in North China during wheat burning season 2014

Yanhong Zhu [1], Lingxiao Yang [1,7*], Jianmin Chen [1,6,7], Kimitaka Kawamura [3,a], Mamiko Sato [3], Andreas Tilgner [4], Dominik van Pinxteren [4], Ying Chen [4,b], Likun Xue [1], Xinfeng Wang [1], Isobel J. Simpson [5], Hartmut Herrmann [4,2,1], Donald R. Blake [5], Wenxing Wang [1]

[1] Environment Research Institute, Shandong University, 250100 Jinan, China

[2] School of Environmental Science and Engineering, Shandong University, Jinan 250100, China

[3] Institute of Low Temperature Science, Hokkaido University, Sapporo 060-0819, Japan

[a] Now at: Chubu Institute of Advanced Studies, Chubu University, Kasugai 487-8501, Japan

[4] Leibniz Institute for Tropospheric Research (TROPOS), 04318 Leipzig, Germany

[b] now at: Lancaster Environment Centre, Lancaster University, Lancaster LA1 4YQ, UK

[5] Department of Chemistry, University of California at Irvine, Irvine, CA, USA

[6] Shanghai Key Laboratory of Atmospheric Particle Pollution and Prevention (LAP3), Fudan Tyndall Centre, Department of Environmental Science and Engineering, Fudan University, Shanghai 200433, China

[7] Jiangsu Collaborative Innovation Center for Climate Change, China

*To whom correspondence should be addressed: Lingxiao Yang: yanglingxiao@sdu.edu.cn

## Abstract.

Fine particulate matter (PM$_{2.5}$) samples collected at Mount Tai in the North China Plain during summer 2014 were analyzed for dicarboxylic acids and related

compounds (oxocarboxylic acids and α-dicarbonyls) (DCRCs). The total concentration of DCRCs was $1050 \pm 580$ ng m$^{-3}$ and $1040 \pm 490$ ng m$^{-3}$ during the day and night, respectively. Although these concentrations were about 2 times lower than similar measurements in 2006, the concentrations reported here were about 1-13 times higher than previous measurements in other major cities in the world. Molecular distributions of DCRCs revealed that oxalic acid ($C_2$) was the dominant species (50%), followed by succinic acid ($C_4$) (12%) and malonic acid ($C_3$) (8%). WRF modeling revealed that Mt. Tai was mostly in the free troposphere during the campaign and long-range transport was a major factor governing the distributions of the measured compounds at Mt. Tai. A majority of the samples (79%) had comparable concentrations during the day and night, with their day-night concentration ratios between 0.9-1.1. Multi-day transport was considered as important reason for the similar concentrations. Correlation analyses of DCRCs and their gas precursors and between $C_2$ and sulfate indicated precursor emissions and aqueous phase oxidations during long-range transport also likely play an important role, especially during the night. Source identification indicated that anthropogenic activities followed by photochemical aging accounted for about 60% of the total variance and was the dominant source at Mt. Tai. However, biomass burning was only important during the first half of measurement period. Measurements of potassium ($K^+$) and DCRCs were about 2 times higher than those from the second half of measurement period. The concentration of levoglucosan, a biomass burning tracer, decreased by about 80% between 2006 and 2014, indicating that biomass burning may have decreased between 2006 and 2014.

## 1 Introduction

Fine particulate matter (PM$_{2.5}$) is an atmospheric pollutant of particular concern due to its contribution to visibility degradation (Ghim et al., 2005; Watson, 2002), exacerbation of respiratory diseases (Davidson et al., 2005) and modification of climate (Sloane et al., 1991). In recent years, haze frequently occurs in China and has

received increasing attention due to its serious impact on air quality and human health (Mu and Zhang, 2013; Guo et al., 2014; Wang et al., 2014). Previous studies have demonstrated that $PM_{2.5}$ is a major pollutant causing haze, particularly its secondary components (Huang et al., 2014). Dicarboxylic acids and related compounds (oxocarboxylic acids and α-dicarbonyls) (DCRCs) are important constituents in $PM_{2.5}$, and mainly produced by secondary processes (Kawamura and Yasui, 2005; Pavuluri et al., 2010a). Due to their high water solubility, DCRCs contribute to the water soluble organic fraction of $PM_{2.5}$, which can have an impact on air quality (van Pinxteren et al., 2009; Kawamura and Bikkina, 2016; Kundu et al., 2010b). Therefore, it is necessary to study the DCRC characteristics in $PM_{2.5}$.

The North China Plain (NCP) is one of the most heavily polluted regions in China, and possibly in the world (Ohara et al., 2007). The region is characterized by high loading of DCRCs due to its high emissions of primary (such as fossil fuel and biomass combustion) and secondary (atmospheric oxidations of biogenic and anthropogenic volatile organic compounds (VOCs)) sources of DCRCs (Kundu et al., 2010a; Mkoma et al., 2013). A previous study has reported that the NCP is an important coal consumer (Xu, 2001). In addition, the NCP is one of the most productive agricultural regions in China, and agricultural waste burning occurs frequently during the harvest seasons. Although some management strategies have been implemented by the Chinese government, such as lawful punishment or punishment by a fine, biomass burning still occurs during the harvest seasons (Zhu et al., 2017). Moreover, the NCP is also one of the highest VOC emission regions in China (Zhang et al., 2009). In order to comprehensively understand the atmospheric formation and processing of DCRCs in the NCP, knowledge of their characteristics over remote background areas is necessary. Moreover, understanding aerosol pollution characteristics over remote background regions is central to identify source regions and the impact of long-range transport.

In this study, $PM_{2.5}$ samples were collected at the top of Mount Tai (Mt. Tai) in the NCP during wheat burning season. As the highest mountain in the NCP, Mt. Tai provides an ideal site to investigate long-range transport. There are many tourists at

Mt. Tai in summer, so there are some local emissions from small restaurants and temples (Gao et al., 2005). Furthermore, 80% of Mt. Tai is covered by vegetation (mostly bushes). Moreover, it should be noted that mountain areas, with parallel ridges or isolated ridges and peaks, are different from the plain in terms of geometric structures. This has implications for modifying the ambient air flow by this complex terrain, which leads to complexity of the mountain boundary layer structure (Smith et al., 2002). Naturally, the boundary layer structure plays important roles in the transport and dispersal of atmospheric pollutants during long-range transport (Garratt, 1994).

The objectives of this study were: (1) to identify the impact of long-range transport by WRF modeling and back-trajectory analysis, (2) to investigate the measured concentrations and compositional trends of DCRCs, (3) to compare our measurements with previous studies at Mt. Tai and other locations, (4) to study the diurnal trend of the compounds, (5) to characterize the biomass burning impact on temporal variations of DCRCs, and (6) to identify potential sources of DCRCs using principal component analysis (PCA).

## 2 Experimental methods

In our previous publication (Zhu et al., 2017), we described the meteorological conditions, sampling site, $PM_{2.5}$ sampling, VOC sampling and analysis from 04 June to 04 July 2014 at Mt. Tai. Therefore, in this study we describe these experimental methods only briefly.

### 2.1 Sampling site for $PM_{2.5}$

$PM_{2.5}$ sampling was conducted at the top of Mt. Tai (36.25N, 117.10E, ~1532.7 m a.s.l.). Mt. Tai is located in the Shandong province in the NCP in a deciduous forest zone and is surrounded by urban and industrialized regions (Gao et al., 2005; Richter et al., 2005). The meteorological data during the sampling period are summarized in Fig. 1. The ambient temperatures covered a range of 10-25 ℃ with an average of

17 ℃. Relative humidity (RH) varied between 58 to 100% with an average of 87%. Winds generally came from the northwest, and wind speeds ranged from 1-7 m s$^{-1}$. Weather conditions during the campaign were mostly cloudy and occasionally foggy. Minor rain events occurred on 15, 16 June and 3 July, and major rain events occurred on 24 June and 4 July. The sample collection was ended just before the major rain.

The PM$_{2.5}$ sampler was placed at the Air Force Hotel in the Houshiwu area. The region is not typically frequented by tourists and is not near any temples. Using a TH-16A Intelligent PM$_{2.5}$ sampler (Wuhan Tianhong Corporation, China) and quartz fiber filters, PM$_{2.5}$ was sampled at 100 L min$^{-1}$. In order to identify the impact of atmospheric chemistry processes on DCRCs, PM$_{2.5}$ samples were collected during the day and night, respectively, from 4 June to 4 July 2014. The time of sunrise and sunset in June at Mt. Tai was around 06:00 and 18:00, respectively. Therefore, 06:00-18:00 and 18:00-06:00 local time have been selected as the sampling times for day and night, respectively. Before each sample, the filters were pre-heated at 600 ℃ for 4 h. Blank samples were collected between 06:00-18:00 and 18:00-06:00 local time from 5-7 July 2014, and their sampling manner was similar to the real samples, but without pumping. After the samples were collected, loaded filters and blank samples were stored in plastic petri dishes and transported to the laboratory, where they were stored at -20 ℃.

**2.2 VOC sampling**

VOC samples were collected by stainless steel canisters from 04 June to 04 July 2014. They were instantaneously sampled and the sampling times were 8:00, 14:00 and 0:00. A total of 70 VOC samples were collected. After sampling, the canisters were shipped to the University of California, Irvine for further analysis. VOC samples were identified and quantified by gas chromatography equipped with electron capture detection (ECD), flame ionization detection (FID) and mass spectrometer detection (MSD). Detailed descriptions about the chemical analysis have been presented in Blake et al. (1994) and Simpson et al. (2010).

**2.3 WRF Model**

The boundary layer heights (BLHs) around Mt. Tai during the campaign were calculated using Weather Research and Forecasting Model (WRF V3.5.1) (Skamarock et al., 2005; Wang et al., 2007). In this study, the Yonsei University (YSU) boundary layer scheme (Hong et al., 2006) was used. In YSU scheme, the boundary layer approaches its top when the critical bulk Richardson number is zero (Hong et al., 2006). The YSU scheme simulates deep vertical mixing accurately in buoyancy driven BLHs (Hong et al., 2006), and reasonably captures the diurnal cycle of BLHs and thermodynamic vertical structure of atmosphere (Hu et al., 2010). The daytime boundary layer structure is well represented by the WRF model with the YSU BLH scheme (Hu et al., 2013). Previous studies also reported that the WRF model can capture the boundary layer structure and local circulation over the NCP mountainous region during summer (Chen et al., 2009), and well captures the vertical structure of potential temperature (Hu et al., 2014). Although the model may have lower confidence for the night BLH estimation (Hu et al., 2013), the larger uncertainty at night would not significantly influence our analysis and conclusions because the nighttime boundary layer was always lower than the measurement site. The reference height of the calculated BLH was sea level. More details of the model configurations were given in Chen et al. (2016).

**2.4 Analytical procedures**

Aliquots of filter samples were extracted by Milli Q water under ultrasonications. The extracts were concentrated by a rotary evaporator. The concentrates were reacted with 14% $BF_3$/n-butanol to convert to dibutyl esters and butoxy acetals. The derivatives were dissolved in n-hexane and analyzed by an Agilent 6890 gas chromatograph (GC) installed with a split/splitless injector, fused silica capillary column (HP-5, 0.2 mm $\times$ 25 m, film thickness 0.5 μm) and a flame ionization detector (FID). Identification of each compound was based on a comparison of retention times of GC peaks with those of authentic standards and confirmed by GC/mass spectrometry. Detection limits of

the measured chemical species were 0.05 to 0.1 ng m$^{-3}$, which were calculated on the basis of minimum areas. Recoveries were 85% for oxalic acid (C$_2$), 90% for malonic acid (C$_3$) and more than 90% for succinic (C$_4$), glutaric (C$_5$) and adipic (C$_6$) acids. Although field blanks revealed peaks of C$_2$ and phthalic acid (Ph), the concentrations were below 5% of the real sample concentrations. The data reported in this study have been corrected for the field blanks. The analytical errors based on duplicate analysis were below 10%. Overall uncertainties for DCRC species were about 15% (see Boreddy et al., 2017 for details). A more detailed description of the analytical method was given in previous reports (Kawamura and Ikushima, 1993; Hegde and Kawamura, 2012).

Additionally, organic carbon (OC) and elemental carbon (EC) were detected by a Sunset Laboratory carbon analyzer with the thermal-optical transmittance method. Detailed descriptions of this methodology have been presented in Cui et al. (2016). Water-soluble ions were analyzed by an ion chromatography system (ICs-90, Dionex Corporation, USA). Analyses of the ionic components were presented in Zhu et al. (2015).

**2.5 Back-trajectory analysis**

72 hour back-trajectories were calculated using the Hybrid Single-Particle Lagrangian Integrated Trajectory model (HYSPLIT, version 4.9) (Draxler and Rolph, 2003) and Global Data Assimilation System (GDAS) meteorological data from the NOAA Air Resources Laboratory's web server. Three-dimensional (latitude, longitude and height) backward trajectories were computed every 1 hour from 04 June to 04 July 2014. 744 trajectories were obtained and classified into four different groups. Yuan et al. (2014) have given detailed descriptions about cluster analysis.

**2.6 The principal component analysis (PCA) method**

Principal component analysis (PCA) is a multivariate analytical tool. It starts with a great many correlated variables and attempts to find a smaller number of independent factors, which can explain the variance in data. Here, the compound concentrations

should firstly be transformed into standardized form using the following formula:

$$Z_{ij} = \frac{C_{ij} - \overline{C_j}}{\sigma_j}$$

where $i = 1, \cdots, n$ sample; $j = 1, \cdots, m$ compound; $C_{ij}$ is the concentration of compound $j$ in sample $i$; and $\overline{C_j}$ and $\sigma_j$ are the arithmetic mean concentration and the standard deviation for compound $j$, respectively. The derived variables are linear combinations of original variables (Callén et al., 2009). In order to better identify the influence of the original variables, varimax rotation is used to obtain the rotated factor loadings that reflect the contribution of each variable to its principal component (PC) (Almeida et al. 2005; Viana et al., 2006). Factor loading means the correlation coefficient between the variable and the PC, which reveals how much a variable contributes to the corresponding PC and how much a variable differs from others. Only factors with eigenvalues greater than 1 are extracted based on Kaiser-Meyer-Olkin (KMO) and the Bartlett's test of sphericity.

## 3 Results and discussion

### 3.1 Air mass back-trajectory analysis

To identify the impact of regional transport, we calculated back-trajectories using the HYSPLIT model (Draxler and Rolph, 2003). The mean transport pathway and corresponding total concentration of DCRCs for every cluster are displayed in Fig. 2. Clusters 2 and 4 accounted for 79% of the trajectories. Moreover, the total concentration of DCRCs was the greatest in clusters 2 and 4. The source regions of the air in clusters 2 and 4 were characterized by large emissions of VOCs (Zhang et al., 2009), which are important precursors of DCRCs. As a result, clusters 2 and 4 had higher DCRC concentrations. The sum of DCRC concentrations in clusters 2 and 4 contributed 73% of the total concentration of DCRCs during the sampling period. Sampling dates in cluster 2 included 6 June, 11-13 June, 22-24 June, 28 June and 4 July, while 9-10 June, 14-21 June, 25-27 June, 30 June and 1-3 July belonged to

cluster 4. Clusters 1 and 3 originated from cleaner areas (i.e., the ocean and Siberia, respectively), so the total concentration of DCRCs was lower compared with clusters 2 and 4. Trajectories on 4-5 June and 29 June were grouped into cluster 1, while trajectories on 7-8 June were grouped into cluster 3. Using WRF modeling, the BLHs at Mt. Tai (Fig. 3) were calculated. The results revealed that the day BLHs were just occasionally higher than the site elevation for only 5% of cluster 2 and 9% of cluster 4 trajectories, while the night BLHs were all lower than the site elevation. Mixed layer heights in the back-trajectory clusters (Fig. S1) (GDAS1, Air Resources Laboratory, NOAA) were all lower than the sampling site elevation. Therefore, our measurements generally represented concentrations in the free troposphere, which suggested that pollutant concentrations at Mt. Tai were largely controlled by long-range transport. The result was different from a previous study at Mt. Tai in 2006 (Kanaya et al., 2013), which reported that the daytime BLHs during the campaign were mostly higher than the observation site, and the night BLHs were generally within the residual layer and occasionally in the free troposphere. Different meteorological conditions between 2006 and 2014 at Mt. Tai, such as higher air pressure conditions but quite low wind speeds in 2014, were the probably reasons for the BLH differences.

**3.2 Measured concentrations of DCRCs and their compositional trends**

Homologous series of dicarboxylic acids ($C_2$-$C_{12}$), oxocarboxylic acids ($C_2$-$C_9$ except $C_6$) and α-dicarbonyls ($C_2$-$C_3$) were detected in $PM_{2.5}$ samples during the day and night at Mt. Tai (Table 1). The total concentration of all detected DCRCs was 1050 ± 580 ng m$^{-3}$ during the day and 1040 ± 490 ng m$^{-3}$ during the night. According to back-trajectories and classification results of DCRCs in different back-trajectory clusters in Fig. 2, we can see that DCRC concentrations were mostly higher in air masses that originated from north of Mt. Tai (northern Hebei province) (31%) and south of Mt. Tai (northern Anhui province) (48%), but lower in air masses derived from the ocean (11%) and Siberia (10%). $PM_{2.5}$ mass concentration at Mt. Tai during the campaign was 98.2 ± 29.2 and 98.6 ± 25.3 μg m$^{-3}$ during the day and night, respectively. DCRCs total concentration contributed about 1.2% and 1.1% to $PM_{2.5}$ in

the day and night, respectively. In addition, the DCRCs-C accounted for 3.3% and 3.2% of OC in the day and night, respectively.

As shown in Table 1, the concentrations of DCRC individual species were comparable during the day and night. $C_2$ was found to be the most abundant dicarboxylic acid compound. The relative abundance of $C_2$ in the total concentration of dicarboxylic acids was 57% during the day and 60% during the night, followed by $C_4$ (day: 14%, night: 14%) and $C_3$ (day: 10%, night: 9%). These results were consistent with the previous 2006 studies at Mt. Tai (Wang et al., 2009; Kawamura et al., 2013). Ph (day: 4%, night: 3%), $C_5$ (day: 3%, night: 3%) and azelaic acid ($C_9$) (day: 2%, night: 2%) also exhibited some contributions. The other dicarboxylic acid species contributed less than 2%. Among the oxocarboxylic acids, $\omega C_2$ provided the largest contribution to their total concentration (day: 43%, night: 46%), followed by Pyr (day: 18%, night: 16%) and 4-oxobutanoic acid ($\omega C_4$) (day: 13%, night: 13%). Additionally, two α-dicarbonyls were identified (Gly and MGly). During the day they had similar concentrations, but MGly exhibited a higher concentration than Gly during the night.

## 3.3 Comparison with previous aerosol studies at Mt. Tai in 2006 and other urban sites in the world

The concentrations of DCRCs at Mt. Tai in 2014 and from other previous measurements are presented in Table 2. Deng et al. (2011) reported that the ratio of $PM_{2.5}$/TSP in June 2006 at Mt. Tai was 0.91. Deng et al. (2011) also showed that most of the water-soluble ions presented similar concentrations in $PM_{2.5}$ and TSP, and the ratios of their concentrations in $PM_{2.5}$ and TSP were more than 0.9. Therefore, we assumed there were small contributions of DCRCs from coarse mode particles. The low impact of particle size on particle composition has been reported at Mt. Gongga in China (Yang et al., 2009). Using the ratio of $PM_{2.5}$/TSP ($PM_{2.5}$/TSP = 0.91) and DCRC concentrations in TSP at Mt. Tai in June 2006 (Kawamura et al., 2013), we have estimated the corresponding DCRC concentrations in $PM_{2.5}$ at Mt. Tai in June 2006 (1550, 220, 62 ng m$^{-3}$ for dicarboxylic acids, oxocarboxylic acids and

α-dicarbonyls, respectively).

Compared to the results from this study, DCRC concentrations in 2014 at Mt. Tai were about two times lower. Different meteorology conditions in 2006 and 2014 may partially explain the decreased concentrations, as well as the implementation of regulatory controls of biomass burning by the Chinese government. In addition, using the ratio of $PM_{2.5}$/TSP and the levoglucosan concentration in TSP at Mt. Tai in June 2006 (Fu et al., 2008), the estimated levoglucosan concentration in $PM_{2.5}$ at Mt. Tai in June 2006 was 390 ng m$^{-3}$. The result was more than five times higher than that in 2014 (levoglucosan: 70 ng m$^{-3}$) (Zhu et al., 2017), which suggests that biomass burning may have decreased from 2006 to 2014, or Mt. Tai was less influenced by emissions from lower altitudes during summer 2014.

Compared with the Chinese megacities, such as Guangzhou in 2007 and Beijing in 2006 (Ho et al., 2010; Ho et al., 2011), the total concentration of DCRCs at Mt. Tai in 2014 was about 1-2 times higher. The concentration of dicarboxylic acids at Mt. Tai in 2014 was similar to the concentration reported in 14 Chinese cities in 2003 (Ho et al., 2007), while oxocarboxylic acids and α-dicarbonyls were more than three times higher at Mt. Tai. Compared with other Asian urban sites, the total concentration of DCRCs reported here was about 1-2 times higher when compared with those reported in $PM_{10}$ in Chennai, India in 2007 (Pavuluri et al., 2010b), and in TSP in Tokyo, Japan in 1989 (Kawamura and Yasui, 2005) and Sapporo, Japan in 2005 (Aggarwal and Kawamura, 2008), but lower than that in $PM_{2.1}$ in Raipur, India in 2012-2013 (Deshmukh et al., 2016). Furthermore, the Mt. Tai DCRCs total concentration in 2014 was approximately 13 times higher compared with Houston, USA in 2000 (Yue and Fraser, 2004). Meanwhile, the result reported in this study was about 5 and 13 times higher than those in $PM_{10}$ in Leipzig, Germany in 2003-2005 (van Pinxteren et al., 2014) and in TSP in Zurich, Switzerland in 2002 (Fisseha et al., 2006), respectively. The high DCRC concentrations at Mt. Tai likely resulted from the substantial emissions of VOCs in Mt. Tai's surrounding areas, which are some of the most heavily polluted regions in China.

$C_2$ was the most abundant species at Mt. Tai in 2014, and the concentrations of $C_4$

were larger than $C_3$. This trend was consistent with measurements at several urban sites, where anthropogenic emissions were important sources, such as in the 14 Chinese cities and Tokyo, Japan (Ho et al., 2007; Kawamura and Yasui, 2005). Ph, a tracer for anthropogenic sources (Kawamura and Yasui, 2005), was the second most abundant species in Beijing, Guangzhou and the 14 Chinese cities. However, in this study, Ph was just the fifth most abundant species. The dissimilar trend was likely due to photochemical aging during long-range transport from source regions to Mt. Tai.

Although DCRC concentrations decreased from 2006 to 2014, they were still greater compared to other urban sites in the world. Due to photochemical aging during long-range transport, the compositional trends of DCRCs were different from previously studied Chinese sites.

### 3.4 Comparisons of day and night measurements of DCRCs

As shown in Table 1, the daytime concentrations of individual DCRC species were similar to their nighttime concentrations. The day-night concentration ratios of 79% of the individual species ranged between 0.9 and 1.1.

The BLHs were higher during the day, peaking near noontime. The boundary layer occasionally extended high enough during the day to approach the sampling site (Fig. 3). However, the maximum BLH was only ~ 600 m during the night, which was much lower than the sampling site height. As shown in Fig. 3, the total concentration of DCRCs increased when BLHs were higher than the site elevation, which suggested that mountain/valley breezes may bring ground-level pollutants to the summit of Mt. Tai. During the day when the BLHs can be above the sampling site height, more polluted air can be transported from the lower (ground) levels to Mt. Tai top, while during the night, cooling of the ground surface and subsidence of cool air may pull down clean air masses from the free troposphere to the top of Mt. Tai (Fu et al., 2014). However, clear diurnal variations were not found in the DCRC concentrations. Furthermore, the predicted BLH (Fig. 3) suggests that the sampling site was mostly above the BLH during the sampling period, thus the impact of uplifted air on the 12h filter measurements should be minor. Moreover, it is noted that the summit of Mt. Tai

is about a few hundred meters above other summits in the surrounding region (Fig. 4). Therefore, the airflow at Mt. Tai should be mainly influenced by the synoptic flow rather than drainage flows. Such an isolated mountain peak is often characterized by wind flows around the peak and small amounts of lifting over it. Nevertheless, under light wind conditions, sunlit mountain slopes may be a favored location for thermals lifting up air from lower levels. However, due to the predominant northwesterly winds, this might have only a minor effect on the performed measurements. No day-night variations of the DCRCs were observed, indicating similar air masses throughout the day and night measurement periods. Due to the fact that air masses arriving at Mt. Tai are transported over several days, multi-day transport has to be considered as part of the reason for the similar concentrations of the field samples taken during the day and night.

As shown in Fig. 5, DCRC concentrations exhibited weak and moderate correlations with total the concentration of selected DCRC precursors during the day ($R^2 = 0.29$) and night ($R^2 = 0.48$), respectively, where selected DCRC precursors included ethyne, ethene, isoprene, α-pinene, β-pinene, toluene, m/p-xylene and o-xylene (Warneck, 2003; Ervens et al., 2004; Bikkina et al., 2014; Tilgner and Herrmann, 2010).

In addition, the daytime DCRC concentrations might have been enhanced by photochemical reactions, and the night concentrations might have been enhanced by effective aqueous oxidation and less effective loss. Average RH values during the sampling period at Mt. Tai were 87%, up to 100% (Fig. 1), and higher on average during the night (Fig. 6). In addition, average RH values along the dominant back-trajectory clusters (clusters 2 and 4) were about 70% (Fig. S1). However, due to the coarse resolution of HYSPLIT, it was difficult to judge whether clouds occurred. Therefore, MODIS satellite pictures were investigated, and the results showed that clouds sometimes occurred in the region of Mt. Tai and in the areas that the trajectories passed over during the sampling period. But MODIS satellite pictures have limited information about the cloud base and top heights, and thus it cannot exactly explore whether there were clouds at the height of the trajectories. A

correlation of $C_2$ and sulfate ($SO_4^{2-}$) and the corresponding linear regression slope were used to evaluate whether $C_2$ was produced by aqueous phase oxidation (Yu et al., 2005; Sullivan and Prather, 2007). As shown in Fig. 7, $C_2$ and $SO_4^{2-}$ exhibited a higher correlation during the night ($R^2 = 0.64$) than that during the day ($R^2 = 0.28$), and the linear regression slope during the night (0.028) was also higher than that during the day (0.016). Assuming aqueous phase formation of $SO_4^{2-}$ was the dominant process (Yu et al., 2005), these results indicated that a substantial concentration of $C_2$ may be produced via aqueous-phase oxidation during the night. In addition, photolysis of iron-oxalate complexes is considered as an important sink of $C_2$, which is effective under clear-sky sunlight conditions (Ervens et al., 2003; Pavuluri and Kawamura, 2012; Weller et al., 2014). Deng et al. (2011) and Shen et al. (2012) reported that Mt. Tai aerosol particles and cloud droplets include a substantial amount of transition metal ions, such as iron. Deng et al. (2011) reported that iron concentration was 0.71 μg m$^{-3}$ in PM$_{2.5}$ and 1.69 μg m$^{-3}$ in TSP during summer 2006. Moreover, Shen et al. (2012) reported that the average bulk cloud water concentration of iron was 44 μg L$^{-1}$ and 416 μg L$^{-1}$ during summer 2007 and 2008, respectively. Thus, iron-oxalate complex formation and photolysis might be possible chemical pathways occurring in Mt. Tai aerosols. Therefore, the removal of $C_2$ was lower during the night than during the day.

Details of the multiphase formation pathways, removal mechanisms and major precursor contributions will be further investigated using the SPectral Aerosol Cloud Chemistry Interaction Model (SPACCIM, Wolke et al., 2005) together with chemical aqueous-phase radical mechanism (CAPRAM, Tilgner et al., 2013) in an upcoming study.

**3.5 Impact of biomass burning on the temporal variations of DCRCs**

The temporal variations of DCRCs and K$^+$ are presented in Fig. 8. It can be seen from Fig. 8 that DCRC concentrations in the first half of the sampling period (4-19 June) were about two times higher than those in the second half of the sampling period (20 June-4 July). From the trajectory analysis, we can see that during the first and second

half of the sampling periods, 4 and 5 days, respectively, belonged to cluster 2. In addition, 8 and 9 days belonged to cluster 4, respectively. Therefore, the dominant air masses in the first and second half of the sampling periods were similar, and thus had a low impact on DCRC concentrations in the two periods. Figure S2 shows meteorological data in the different backward trajectory clusters during the sampling period at Mt. Tai. We can see that the pressure, temperature and RH didn't change much in each of cluster 2 and cluster 4 over the timescale of the mean trajectories. Moreover, as shown in Fig. 1, meteorological data at Mt. Tai site also didn't change much between the first and second half of the sampling periods. Therefore, the quite stable meteorological conditions may have had a low impact on the DCRC concentrations between the first and second half of the sampling periods. Dicarboxylic acids and $K^+$ exhibited a strong correlation during the first half of the measurement ($R^2 = 0.77$), while during the second half, dicarboxylic acids and $K^+$ exhibited no correlation ($R^2 = 0.04$) (Fig. 9). The peaks of dicarboxylic acids and $K^+$ appeared almost simultaneously (Fig. 8). It is also clear that when the $K^+$ concentration increased, dicarboxylic acids correspondingly increased during the first half (Fig. 9). These results imply that biomass burning was an important contributor to DCRCs during the first half of the measurement period. Moreover, according to reports by weather satellites of the Ministry of Environment Protection of the People's Republic of China (http://www.zhb.gov.cn/), straw burning hotspots in air masses that passed over key areas (Anhui, Hebei and Shandong province) were mainly distributed in the first half of the sampling period (Fig. 10). This result further supports that biomass burning was more important in the first half of the sampling period.

Concentrations of $C_2$ declined from the first half to the second half (Fig. S2). $C_3$, $C_4$ and longer-chain dicarboxylic acids ($C_5$-$C_9$) also exhibited similar trends, with much higher concentrations during the first half. However, $iC_4$ and $iC_6$ were generally constant throughout the whole period. This result suggested that biomass burning may be an insignificant source for these two species. M, F, mM, Ph, iPh, tPh, $kC_3$ and $kC_7$ had higher concentrations during the first half, suggesting that anthropogenic components, such as vehicle emissions, fossil combustion and plastic burning

(Kawamura and Kaplan, 1987; Simoneit et al., 2005), were probably transported to the Mt. Tai site concurrently with biomass burning plumes.

Oxocarboxylic acids and α-dicarbonyls exhibited similar temporal trends with dicarboxylic acids. On average, oxocarboxylic acids and α-dicarbonyls were more abundant during the first half ($158 \pm 101$ and $32.9 \pm 25.5$ ng m$^{-3}$, respectively) than during the second half ($89.2 \pm 25.1$ and $16.8 \pm 8.82$ ng m$^{-3}$, respectively). $\omega C_2$-$\omega C_5$, $\omega C_9$, Pyr, Gly and MGly also displayed similar trends with $C_2$, and had higher concentrations during the first half.

Ph is a photo-degradation product of anthropogenic aromatic hydrocarbons, and $C_9$ is a photo-oxidation product of biogenic unsaturated fatty acids (Schauer et al., 2002; Kawamura and Ikushima, 1993). As a result, the Ph/$C_9$ ratio was considered as a proxy to evaluate source strength of anthropogenic versus biogenic emissions (Kawamura and Yasui, 2005). In this study, Ph/$C_9$ ratios ranged from 0.32 to 8.64 (average: 3.20) and were mostly higher than 1 (Fig. 11), which were comparable to those from the 14 Chinese cities (Ho et al., 2007) (average in the summer: 3.37), but much higher than in Nanjing, China (average in summer: 1.98) (Wang et al., 2002) and Chennai, India (average in summer: 0.69) (Pavuluri et al. 2010b). These comparisons suggest that anthropogenic sources contributed more significantly than biogenic sources at Mt. Tai. Moreover, the Ph/$C_9$ ratios were higher during the day (range: 0.32-8.64, average: 3.75) compared to those during the night (range: 0.54-7.09, average: 2.53). In addition, the Ph/$C_9$ ratios were higher during the second half when almost no straw burning hotspots were observed.

**3.6 Source identification of DCRCs**

In this study, PCA was employed to identify the DCRC sources in PM$_{2.5}$. Concentrations of $C_2$, $C_3$, $C_4$, $C_5$, $C_6$, $C_9$, $iC_5$, $hC_4$, M, F, Ph, tPh, $kC_3$, Pyr, $\omega C_2$, $\omega C_4$, Gly, MGly (compound abbreviation in Table 1), OC, EC, Na$^+$, NH$_4^+$, K$^+$, NO$_3^-$ and SO$_4^{2-}$ as well as mean trajectory length, solar flux along the trajectory and mixing depth along the trajectory were used for PCA using IBM SPSS Statistics 21.0, and the results are presented in Table 3 and Table 4. If the compound concentration was below

the detection limit, the data were replaced by a value half of the corresponding detection limit (Wold et al., 1987). Only factor loadings $|x| > 0.2$ were considered, and $|x| > 0.6$ were considered high loading and are depicted in bold.

From the daytime samples, five PCs were extracted, and PC1, PC2, PC3, PC4 and PC5 explained 64, 9, 7, 6 and 4%, respectively, of the total variance (90% in total). As shown in Table 3, PC1 was dominated by high loadings of $C_2$-$C_6$, $iC_5$, F, $hC_4$, Ph, $kC_3$, Pyr, $\omega C_2$, $\omega C_4$, Gly, MGly, OC and $K^+$, which were associated with anthropogenic activities (such as agricultural activities) followed by photochemical aging. As mentioned above, Ph and $K^+$ implied anthropogenic sources and biomass burning, respectively. The negative loading of mean trajectory length and mixing depth along the trajectory to PC1 suggested high residence times of trajectories above continental areas. $NH_4^+$, $NO_3^-$ and $SO_4^{2-}$ were dominant species in secondary inorganic aerosols. Their positive correlations with PC2 indicated that PC2 was derived from secondary sources, including gas and aqueous phase chemistry. The low loading of solar flux in PC2 indicated that secondary sources were not primarily driven by radiation. PC3 was enriched in M, F and EC, and was assumed to represent fuel combustion. EC was used as source tracer for fuel combustion (Puxbaum et al., 2007; Zhang et al., 2008). M and F can be emitted from fuel combustion (Jung et al., 2010). In PC4, $C_9$ and $Na^+$ were dominant. Moreover, high correlation was found between $C_9$ and $Na^+$ ($R^2 = 0.69$). These results suggested photooxidation of unsaturated fatty acids emitted from the sea surface together with sea salt was dominant in PC4. $Na^+$ is a tracer for sea salt (Wagenbach et al., 1998). For PC5, tPh was dominant, indicating important contribution by waste burning. tPh can be produced by solid wastes and/or plastic polymer burning (Simoneit et al., 2005; Kawamura and Pavuluri, 2010).

During the night period, four PCs were extracted. PC1, PC2, PC3 and PC4 explained 56, 14, 13 and 6%, respectively, of the total variance (89% in total). As shown in Table 4, $C_2$, $C_6$, $hC_4$, Ph, Pyr, $\omega C_2$, $\omega C_4$, Gly, OC and $K^+$ strongly correlated with PC1, which were attributed to anthropogenic activities followed by photochemical aging. In contrast, $C_3$-$C_5$, $iC_5$, M, F, $kC_3$ and EC displayed strong correlations with PC2. As a result, PC2 was attributed to emissions from fuel

combustion and photochemical reaction. For PC3, MGly, $NH_4^+$, $NO_3^-$ and $SO_4^{2-}$ dominated, suggesting that secondary processing was an important source. Moreover, the contribution of this source to the variance was higher during the night (13%) than that during the day (9%) suggesting that secondary processing was more important during the night. The negative loading of mean trajectory length and mixing depth along the trajectory in PC1, PC2 and PC3 indicates long residence times above the continental areas. The correlation between $C_9$ and $Na^+$ ($R^2 = 0.51$) suggests photooxidation of unsaturated fatty acids emitted from sea surface together with sea salt. As mentioned above, tPh is produced by wastes burning. High correlations of $C_9$, tPh, OC and $Na^+$ in PC4 may reveal a mixed aerosol source related to waste burning and photooxidation of unsaturated fatty acids emitted from the sea surface together with sea salt.

Day and night sources of DCRCs were similar, but there were some differences in source order and contribution. Anthropogenic activities followed by photochemical aging had a higher contribution during the day, which was probably related with higher BLHs during the day. Fuel combustion was the second most important source during the night, and its contribution was also higher during the night. Although secondary processing was the third most important source at night, its contribution was higher than that during the day, which may be related to more effective aqueous oxidation during the night. The daytime sources in PC4 and PC5 were not separated during the night.

## Conclusions

Dicarboxylic acids and related compounds (DCRCs) were quantified in $PM_{2.5}$ filter samples collected between 04 June and 04 July 2014 at Mt. Tai. DCRC concentrations were higher than those at urban sites around the world but lower than previous measurements at Mt. Tai. WRF modeling and back-trajectory analysis implied that long-range transport of pollutants was a major factor governing the DCRC distributions at Mt. Tai. PCA results revealed that anthropogenic activities followed

by photochemical aging were the major source for DCRCs at Mt. Tai. Biomass burning only had an important impact in the first half of the measurement period (4-19 June).

Campaign-averaged DCRC concentrations were similar during the day and night. Multi-day transport of pollutants over Mt. Tai was considered as an important factor for the similar concentrations. Based on the correlation analysis between DCRCs and their gas precursors and the correlation between $C_2$ and sulfate, the similar day-night ratios were probably dependent on precursor emissions and aqueous oxidations. Further interpretations of the complex Mt. Tai dataset using a detailed multiphase chemistry air parcel model will be completed in a follow-up study.

## Acknowledgements

The authors acknowledge the financial support from the National Natural Science Foundation of China (Nos. 21577079) and the Japan Society for the Promotion of Science through Grant-in-Aid (No. 24221001). The authors also acknowledge China Scholarship Council for supporting Yanhong Zhu to study on the project at the Atmospheric Chemistry Department (ACD) of the Leibniz Institute for Tropospheric Research (TROPOS), Germany.

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

**Table 1.** Measured concentrations of DCRCs, PM$_{2.5}$, OC and EC at the top of Mt. Tai from 4 June to 4 July 2014.

| Components, abbreviation | Day (n = 32) | | | | Night (n = 27) | | | | Day-Night Ratio |
|---|---|---|---|---|---|---|---|---|---|
| | Min.[a] | Max.[b] | Mean | SD [c] | Min. | Max. | Mean | SD | Mean |
| I. Dicarboxylic acids (ng m$^{-3}$) | | | | | | | | | |
| Oxalic, C$_2$ | 122 | 1790 | 512 | 304 | 151 | 1280 | 534 | 272 | 1.0 |
| Malonic, C$_3$ | 23.7 | 195 | 86.2 | 33.8 | 17.9 | 141 | 77.4 | 32.6 | 1.1 |
| Succinic, C$_4$ | 25.8 | 485 | 126 | 81.1 | 62.3 | 227 | 121 | 53.7 | 1.0 |
| Glutaric, C$_5$ | 6.9 | 99.3 | 26.3 | 15.9 | 10.5 | 53.7 | 24.9 | 11.3 | 1.1 |
| Adipic, C$_6$ | 4.7 | 46.4 | 12.6 | 7.5 | 4.0 | 31.8 | 12.7 | 7.5 | 1.0 |
| Pimelic, C$_7$ | 0.7 | 22.6 | 4.2 | 4.3 | BDL | 19.1 | 4.4 | 4.1 | 0.9 |
| Suberic, C$_8$ | BDL[d] | 2.9 | 0.3 | 0.7 | BDL | 1.9 | 0.4 | 0.5 | 0.9 |
| Azelaic, C$_9$ | 3.4 | 95.1 | 16.2 | 20.6 | 2.3 | 68.1 | 19.4 | 17.6 | 0.8 |
| Sebabic, C$_{10}$ | BDL | 8.9 | 1.2 | 2.1 | BDL | 13.2 | 1.8 | 3.0 | 0.6 |
| Undecanedioic, C$_{11}$ | BDL | 6.0 | 1.7 | 1.9 | BDL | 7.6 | 1.7 | 2.0 | 1.0 |
| Dodecanedioc, C$_{12}$ | BDL | 1.3 | 0.2 | 0.3 | BDL | 1.8 | 0.3 | 0.5 | 0.7 |
| Methylmalonic, iC$_4$ | BDL | 7.9 | 3.4 | 1.6 | 0.7 | 7.6 | 3.3 | 1.5 | 1.0 |
| Methylsuccinic, iC$_5$ | 2.9 | 32.7 | 8.0 | 5.5 | 3.0 | 18.3 | 8.0 | 4.5 | 1.0 |
| 2-methylglutaric, iC$_6$ | BDL | 6.8 | 1.8 | 1.2 | BDL | 4.2 | 1.6 | 1.1 | 1.1 |
| Maleic, M | BDL | 25.7 | 7.4 | 7.2 | 2.2 | 23.7 | 6.6 | 5.0 | 1.1 |
| Fumaric, F | 1.0 | 15.1 | 3.9 | 2.7 | BDL | 7.1 | 3.2 | 1.9 | 1.2 |
| Methylmaleic, mM | 1.3 | 13.1 | 3.5 | 2.5 | 1.1 | 6.4 | 2.9 | 1.4 | 1.2 |
| Malic, hC$_4$ | 0.2 | 5.2 | 1.2 | 1.0 | 0.3 | 3.3 | 1.4 | 0.8 | 0.9 |
| Phthalic, Ph | 19.3 | 99.4 | 36.8 | 16.1 | 16.3 | 53.9 | 29.3 | 10.2 | 1.3 |
| Isophthalic, iPh | BDL | 13.8 | 1.8 | 2.6 | BDL | 9.1 | 1.6 | 2.2 | 1.1 |
| Terephthalic, tPh | 0.9 | 130 | 13.6 | 24.9 | 0.6 | 155 | 12.9 | 32.2 | 1.1 |
| Oxomalonic, kC$_3$ | 3.7 | 31.5 | 12.6 | 7.0 | 2.1 | 29.5 | 11.3 | 6.8 | 1.1 |
| 4-oxopimelic, kC$_7$ | 2.9 | 29.3 | 11.6 | 6.6 | 2.1 | 27.8 | 11.6 | 5.7 | 1.0 |
| Subtotal | 239 | 2950 | 893 | 479 | 358 | 1970 | 892 | 402 | 1.0 |

| II. Oxocarboxylic acids (ng m$^{-3}$) | | | | | | | | | |
|---|---|---|---|---|---|---|---|---|---|
| Pyruvic, Pyr | 6.3 | 124 | 23.6 | 22.9 | 7.1 | 54.8 | 19.7 | 11.0 | 1.2 |
| Glyoxylic, ωC$_2$ | 8.8 | 241 | 54.7 | 45.2 | 11.9 | 166 | 55.9 | 41.4 | 1.0 |
| 3-oxopropanoic, ωC$_3$ | 2.0 | 24.3 | 8.9 | 5.0 | 1.6 | 24.6 | 8.3 | 4.9 | 1.1 |
| 4-oxobutanoic, ωC$_4$ | 5.5 | 52.4 | 16.0 | 10.1 | 5.5 | 54.2 | 15.3 | 10.6 | 1.0 |
| 5-oxopentanoic, ωC$_5$ | 1.0 | 12.2 | 3.8 | 2.2 | 1.3 | 10.7 | 3.5 | 2.1 | 1.1 |
| 7-oxoheptanoic, ωC$_7$ | 2.1 | 17.8 | 7.3 | 3.3 | 1.9 | 13.7 | 6.7 | 3.0 | 1.1 |
| 8-oxooctanoic, ωC$_8$ | 1.1 | 29.9 | 9.4 | 5.7 | 0.9 | 18.3 | 8.8 | 4.4 | 1.1 |
| 9-oxononanoic, ωC$_9$ | BDL | 10.8 | 3.9 | 3.0 | BDL | 13.2 | 3.6 | 3.7 | 1.1 |
| Subtotal | 26.9 | 496 | 128 | 88.9 | 49.0 | 344 | 122 | 73.6 | 1.0 |
| III.α-dicarbonyls (ng m$^{-3}$) | | | | | | | | | |
| Glyoxal, Gly | 1.8 | 59.6 | 12.3 | 10.3 | 3.2 | 39.3 | 12.3 | 9.7 | 1.0 |
| Methylglyoxal, MGly | BDL | 45.2 | 12.1 | 11.2 | BDL | 59.9 | 13.6 | 13.1 | 0.9 |
| Subtotal | 5.1 | 105 | 24.4 | 20.5 | 8.0 | 94.9 | 25.9 | 21.6 | 0.9 |
| Total (all detected organics) | 271 | 3550 | 1050 | 580 | 429 | 2380 | 1040 | 490 | 1.0 |
| IV. Carbonaceous aerosols (μg m$^{-3}$) | | | | | | | | | |
| PM$_{2.5}$ | 37.0 | 193 | 98.2 | 29.2 | 55.7 | 143 | 98.6 | 25.3 | 1.0 |
| OC | 4.4 | 30.7 | 11.6 | 5.8 | 4.0 | 32.9 | 11.7 | 7.8 | 1.0 |
| EC | 0.5 | 3.3 | 1.3 | 0.7 | 0.4 | 4.7 | 1.5 | 0.9 | 0.9 |

[a] Minimum. [b] Maximum. [c] Standard deviation. [d] BDL: Below detection limit.

**Table 2.** DCRC concentrations reported in this study and literature data from the previous measurements at Mt. Tai in 2006 and other urban sites in the world (unit: ng m$^{-3}$).

| Location | Type | Year | Season | Size | Total dicarboxylic acids | Total oxocarboxylic acids | Total α-dicarbonyls | Major Species |
|---|---|---|---|---|---|---|---|---|
| This study (day) | mountain | 2014 | summer | $PM_{2.5}$ | 893 ±479 | 128 ±88.9 | 24.4 ±20.5 | $C_2 > C_4 > C_3 > \omega C_2$ |
| This study (night) | mountain | 2014 | summer | $PM_{2.5}$ | 892 ±402 | 122 ±73.6 | 25.9 ±21.6 | $C_2 > C_4 > C_3 > \omega C_2$ |
| Mt. Tai, China [a] | mountain | 2006 | summer | TSP | 1702 ±1385 | 242 ±210 | 68.3 ±64.1 | $C_2 > C_4 > C_3 > \omega C_2$ |
| 14 Chinese cities [b] | urban | 2003 | summer | $PM_{2.5}$ | 892 ±457 | 36.7 ±23.7 | 5.2 ±4.1 | $C_2 > Ph > C_4 > C_3$ |
| Guangzhou, China [c] | urban | 2007 | summer | $PM_{2.5}$ | 523 ±134 | 19.5 ±9.6 | 5.1 ±2.1 | $C_2 > Ph > tPh > C_3$ |
| Beijing, China [d] | urban | 2006 | autumn | $PM_{2.5}$ | 760 ±369 | 44.7 ±26.6 | 9.1 ±4.9 | $C_2 > Ph > C_4 > C_9$ |
| Chennai, India [e] | urban | 2007 | summer | $PM_{10}$ | 503 ±118 | 31.7 ±11.2 | 7.1 ±2.0 | $C_2 > tPh > C_3 > C_9$ |
| Raipur, India [f] | urban | 2012-2013 | winter | $PM_{2.1}$ | 1072 | 90.9 | 30.2 | $C_2 > C_4 > C_9 > Ph$ |
| Tokyo, Japan [g] | urban | 1989 | summer | TSP | 726 ±636 | 117 ±95 | 46 ±39 | $C_2 > C_4 > C_3 > Pyr$ |
| Sapporo, Japan [h] | urban | 2005 | summer | TSP | 406 | 35 | 9.7 | $C_2 > C_3 > C_4 > \omega C_2$ |
| Leipzig, Germany [i] | urban | 2003-2005 | summer/winter | $PM_{10}$ | 175 [1] | | | $C_2 > C_3 > C_5 > hC_4$ |
| Zurich, Switzerland [j] | urban | 2002 | summer | TSP | 66.9 [1] | | | $C_2 > C_3 > hC_4 > C_4$ |
| Houston, USA [k] | urban | 2000 | summer | $PM_{2.5}$ | 67.7 [1] | | | $C_4 > C_3 > C_9 > C_5$ |

[a] Kawamura et al. (2013).

[b] Ho et al. (2007).

5    [c] Ho et al. (2011).

[d] Ho et al. (2010).

[e] Pavuluri et al. (2010a).

[f] Deshmukh et al. (2016).

[g] Kawamura and Yasui (2005).

10    [h] Aggarwal and Kawamura (2008).

[i] van Pinxteren et al. (2014).

[j] Fisseha et al. (2006).

[k] Yue and Fraser (2004).

[l] did not include all dicarboxylic acid species.

**Table 3.** PCA factor loadings for daytime DCRCs, OC, EC and inorganic ions as well as mean trajectory length, solar flux along trajectory and mixing depth along trajectory.

| Compounds | PC1[a] | PC2[b] | PC3[c] | PC4[d] | PC5[e] |
|---|---|---|---|---|---|
| $C_2$ | **0.854** | 0.382 | 0.203 | | |
| $C_3$ | **0.832** | 0.277 | | | |
| $C_4$ | **0.751** | 0.353 | 0.407 | | |
| $C_5$ | **0.764** | 0.267 | 0.437 | | |
| $C_6$ | **0.697** | 0.222 | 0.322 | | |
| $C_9$ | | -0.256 | 0.294 | **0.756** | 0.389 |
| $iC_5$ | **0.762** | | 0.523 | | |
| M | | | **0.885** | | |
| F | **0.630** | 0.288 | **0.635** | | |
| $hC_4$ | **0.794** | 0.205 | | | |
| Ph | **0.693** | | 0.431 | | 0.313 |
| tPh | | | | | **0.904** |
| $kC_3$ | **0.716** | | 0.285 | | -0.202 |
| Pyr | **0.823** | 0.353 | 0.218 | | |
| $\omega C_2$ | **0.854** | 0.406 | | | |
| $\omega C_4$ | **0.881** | | | | |
| Gly | **0.834** | 0.396 | 0.248 | | |
| MGly | **0.687** | 0.540 | | | |
| OC | **0.787** | | | 0.559 | |
| EC | 0.411 | 0.226 | **0.632** | | -0.337 |
| $Na^+$ | 0.241 | 0.314 | | **0.862** | |

| | | | | | |
|---|---|---|---|---|---|
| $NH_4^+$ | 0.315 | **0.938** | | | |
| $K^+$ | **0.875** | 0.289 | | 0.293 | |
| $NO_3^-$ | 0.355 | **0.814** | 0.302 | | |
| $SO_4^{2-}$ | 0.279 | **0.895** | | | |
| Mean trajectory length | **-0.629** | **-0.627** | -0.255 | | |
| Solar flux along trajectory | -0.401 | 0.380 | | | |
| Mixing depth along trajectory | -0.507 | 0.393 | -0.302 | | |
| Variance (%) | 64% | 9% | 7% | 6% | 4% |

Extraction method: Principal Component Analysis (PCA).

Rotation method: varimax with Kaiser normalization.

[a] anthropogenic activities followed by photochemical aging

[b] secondary sources

[c] fuel combustion

[d] photooxidation of unsaturated fatty acids emitted from the sea surface together with sea salt

[e] waste burning

**Table 4.** PCA factor loadings for nighttime DCRCs, OC, EC and inorganic ions as well as mean trajectory length, solar flux along trajectory and mixing depth along trajectory.

| Compounds | PC1[a] | PC2[b] | PC3[c] | PC4[d] |
|---|---|---|---|---|
| $C_2$ | **0.674** | 0.504 | 0.464 | |
| $C_3$ | 0.341 | **0.728** | 0.436 | |
| $C_4$ | 0.356 | **0.678** | 0.506 | |
| $C_5$ | 0.578 | **0.699** | 0.285 | |
| $C_6$ | **0.661** | 0.400 | | 0.516 |
| $C_9$ | | 0.531 | | **0.726** |
| $iC_5$ | 0.407 | **0.657** | | 0.585 |
| M | | **0.870** | | 0.239 |
| F | 0.538 | **0.642** | 0.334 | |

| | | | | |
|---|---|---|---|---|
| $hC_4$ | **0.735** | | 0.364 | |
| Ph | **0.610** | 0.478 | 0.305 | 0.467 |
| tPh | | | | **0.953** |
| $kC_3$ | 0.514 | **0.779** | | |
| Pyr | **0.834** | 0.293 | 0.356 | |
| $\omega C_2$ | **0.823** | 0.312 | 0.435 | |
| $\omega C_4$ | **0.893** | 0.261 | | 0.283 |
| Gly | **0.819** | 0.352 | 0.378 | |
| MGly | 0.568 | | **0.671** | |
| OC | **0.674** | 0.223 | | **0.660** |
| EC | | **0.770** | | |
| $Na^+$ | 0.374 | | | **0.865** |
| $NH_4^+$ | 0.273 | | **0.921** | |
| $K^+$ | **0.894** | 0.248 | | |
| $NO_3^-$ | 0.540 | -0.206 | **0.684** | |
| $SO_4^{2-}$ | | 0.365 | **0.887** | |
| Mean trajectory length | -0.564 | -0.408 | -0.531 | |
| Solar flux along trajectory | | | | |
| Mixing depth along trajectory | -0.522 | -0.427 | 0.293 | |
| Variance (%) | 56% | 14% | 13% | 6% |

Extraction method: Principal Component Analysis (PCA).

Rotation method: varimax with Kaiser normalization.

[a] anthropogenic activities followed by photochemical aging

[b] fuel combustion and photochemical reaction

[c] secondary processing

[d] a mixed aerosol source related to waste burning and photooxidation of unsaturated fatty acids emitted from the sea surface together with sea salt

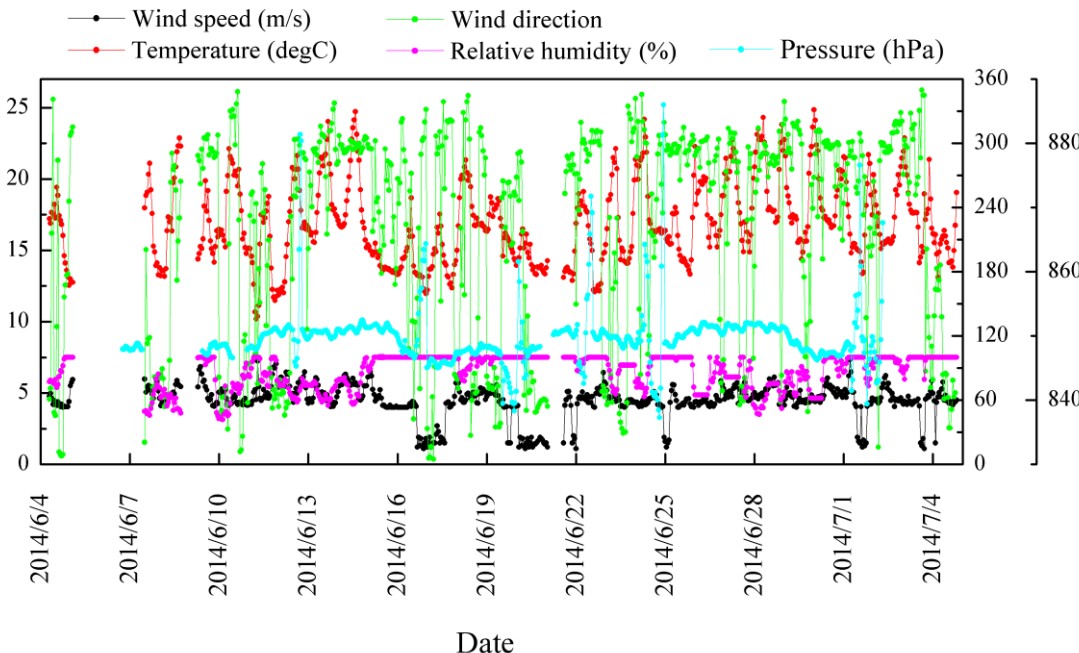

**Fig. 1.** Summary of meteorological data during the sampling period at Mt. Tai. The left y axis is for wind speed and temperature, whereas the right y1 axis (0-360) shows wind direction (degree) and relative humidity, right y2 axis (830-890) shows pressure.

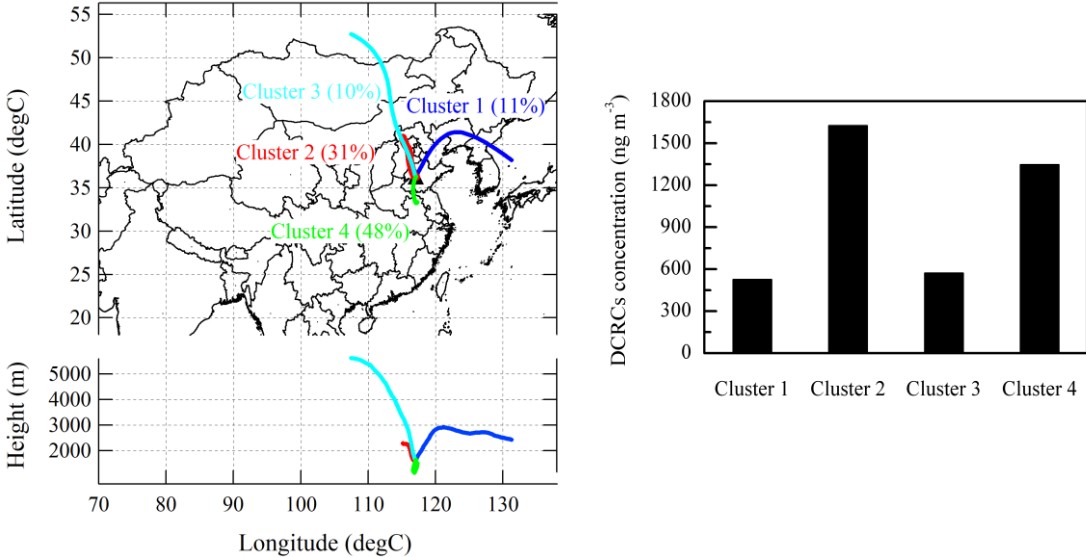

5    **Fig. 2.** Three-day backward trajectories for Mt. Tai during the study period (since cluster 2 was

covered by cluster 3, the width of the cluster 2 has been increased).

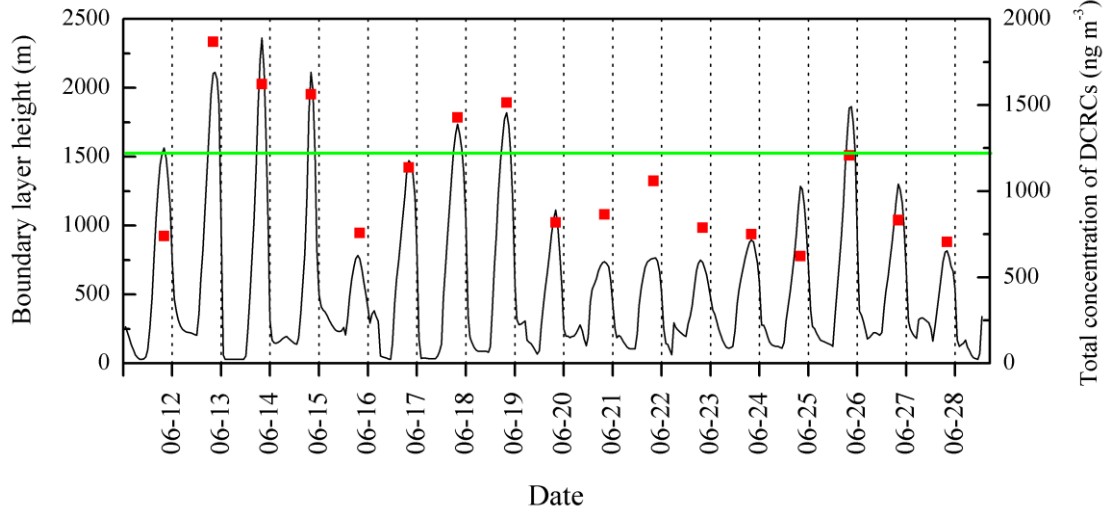

**Fig. 3.** Boundary layer height at Mt. Tai area during selected sampling period modeled with WRF

model (red square: DCRCs concentration; green line: height of sampling site).

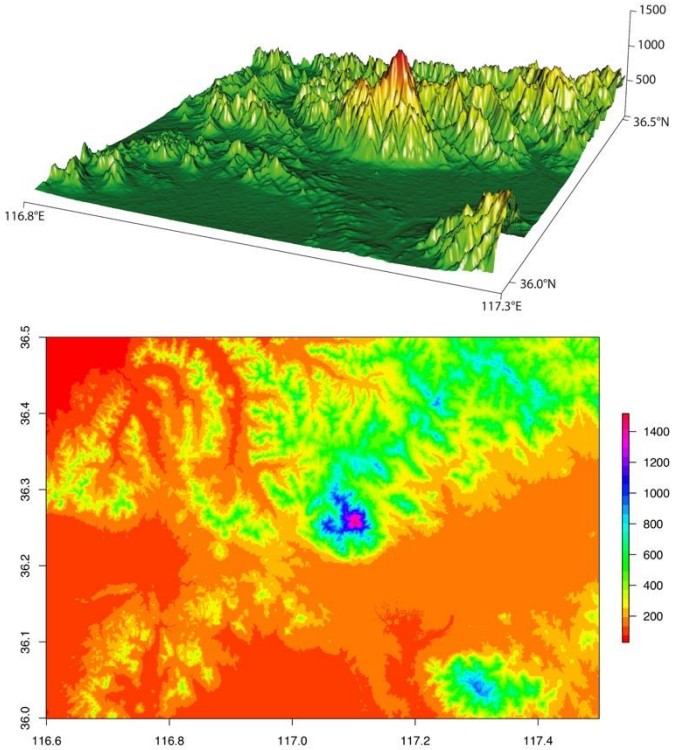

**Fig. 4.** Topographic map of Mt. Tai and the surrounding region. In the top and bottom panels the altitude is shown by the z-axis and by the color-map, respectively, both with units of meters. The digital SRTM (NASA's Shuttle Radar Topography Mission) elevation data are provided by the CIAT-CSI SRTM website (http://srtm.csi.cgiar.org).

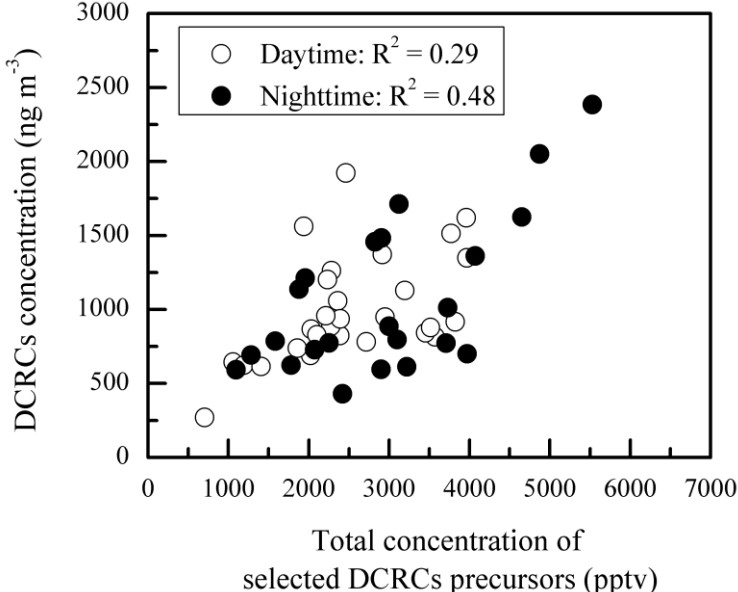

**Fig. 5.** Scatter plot of the day and night concentration of DCRCs and selected DCRC precursors. Here, the total concentration of selected DCRC precursors is the summed concentration of ethyne, ethene, isoprene, α-pinene, β-pinene, toluene, m/p-xylene and o-xylene.

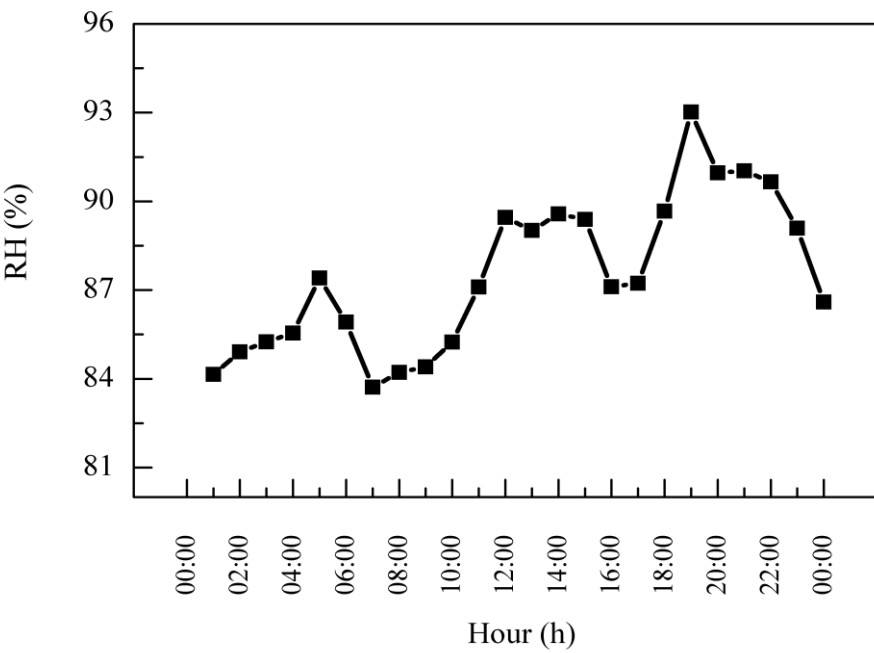

**Fig. 6.** Average diurnal variation of RH during the sampling period at the top of Mt. Tai.

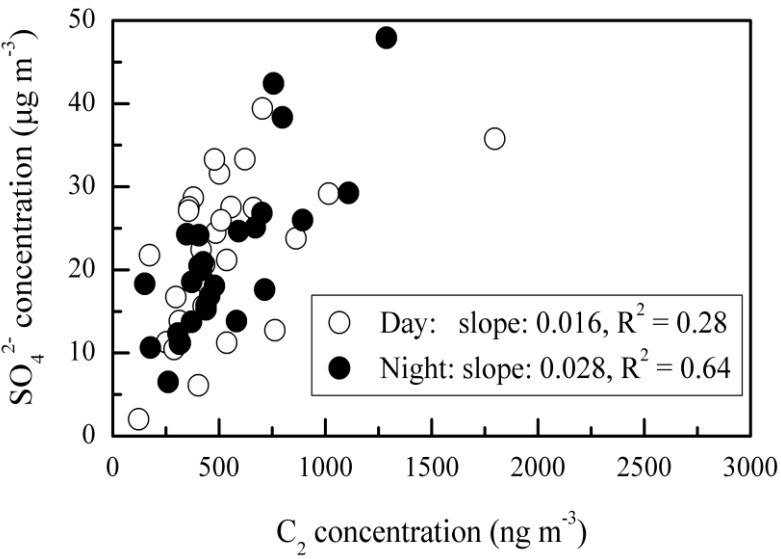

**Fig. 7.** Day and night scatter plot between $C_2$ and $SO_4^{2-}$.

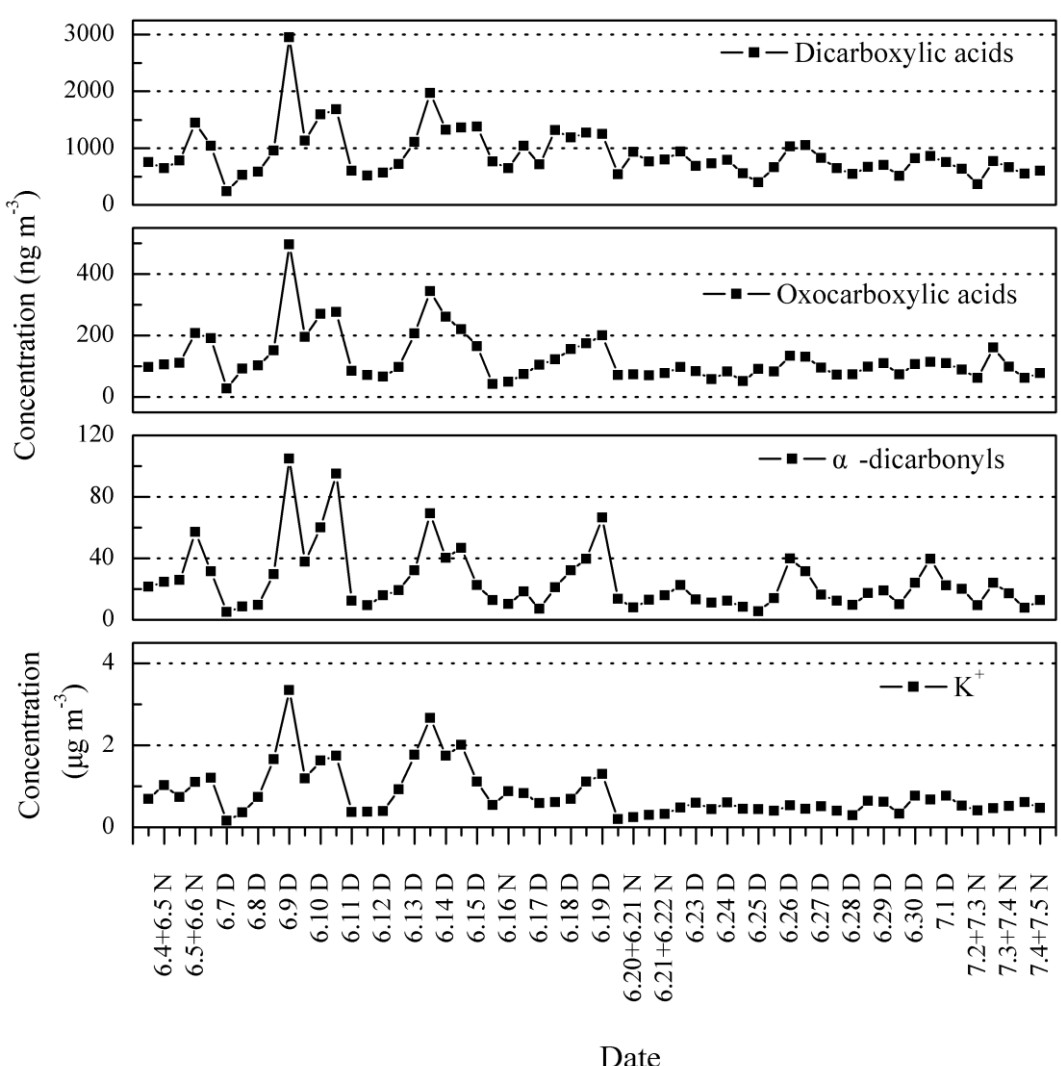

**Fig. 8.** Temporal variation of DCRCs and K$^+$ in PM$_{2.5}$ aerosols collected at Mt. Tai during the day (D) and night (N) in 2014.

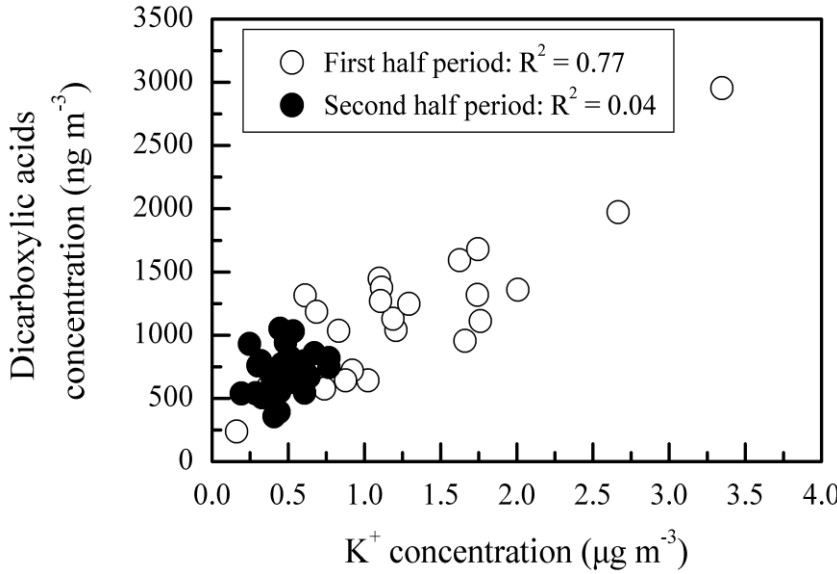

**Fig. 9.** Scatter plot of concentration between $K^+$ and dicarboxylic acids during the first and second half of the campaign.

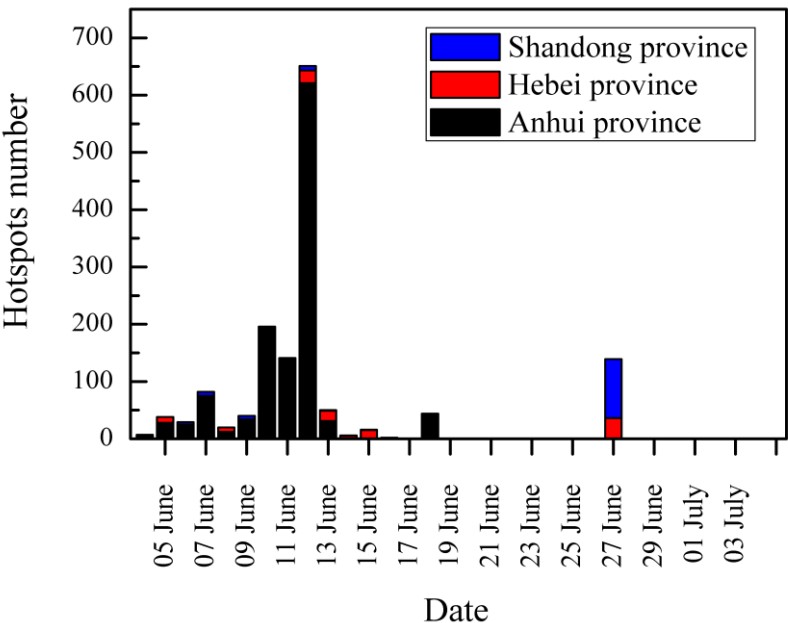

**Fig. 10.** Straw burning hotspots number in air masses passed over key areas during the sampling period reported by weather satellite of the Ministry of Environment Protection of the People's

10   Republic of China.

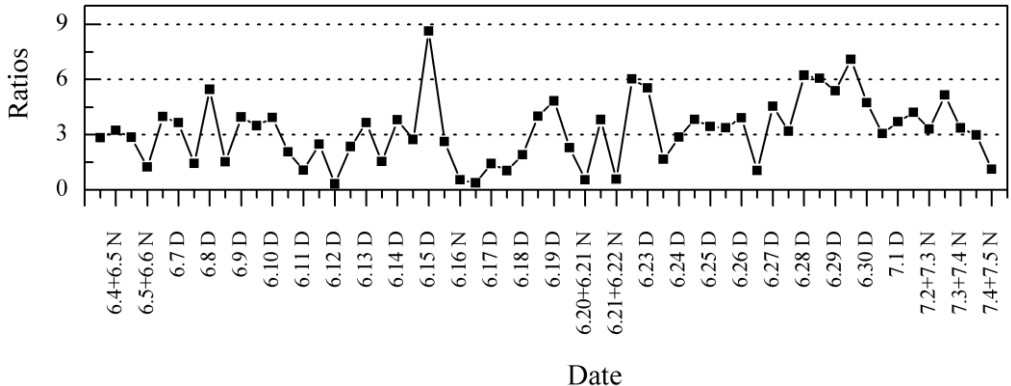

**Fig. 11.** Time series of the Ph/C$_9$ ratios (D: day, N: night).