# Peer review of "Molecular distributions of dicarboxylic acids, oxocarboxylic acids and $\alpha$ -dicarbonyls in $PM_{2.5}$ collected at the top of Mount Tai, in North China during wheat burning season 2014"

_Atmospheric Chemistry and Physics, 2017_

## Referee Comment (RC1) · Anonymous Referee #1 · 21 Feb 2018

The authors present data from measurements at Mt Tai focusing on dicarboxylic acids and related compounds. Based on back trajectory and model analysis they conclude that aerosol arriving on Mt Tai has undergone long range transport and has a variety of sources, including anthropogenic emissions and biomass burning. This study is very similar to previous studies from some of the same authors (27% similarity rate), with some new features (backtrajectories, WRF model) which, however, have not been really made use of. Overall, it may be an interesting data set, in particular as it is discussed in the context of previous measurements of the same compounds at many different locations and also at Mt Tai in 2006. However, I think the discussion is quite confusing and needs major revision.

[Figure]

Major comments

1) Day vs night time samples The authors find that day and night time samples show almost identical concentrations. However, I am not sure that distinguishing day- and night-time-samples is really meaningful here: If a sample was collected at nighttime (i.e. 6 pm - 6 am), it was likely processed during the day(s) before. The same might be true for day time samples that travelled to the sample location for several days. Thus, I am not surprised that samples collected during day and night show very similar composition and loadings. Unless I misunderstood the sampling and nomenclature of day/night samples, I suggest removing the discussion of day- versus night-samples. That way, hypotheses such as on night time oxidation (p. 2, l. 12) or less effective loss during night (p. 10, l. 24) could be removed as they do not seem supported.

2) Trajectories a) In Figure 1, the authors show 72 h-back trajectories of air masses arriving at Mt Tai. These trajectories are briefly discussed in Section 3.1. However, the authors do not link their later discussion to these trajectories. For example, there seems to be change in conditions (meteorology, emissions, air mass?) after the first half of the sampling period that leads to lower diacid loadings. Could that be linked to different trajectories? I suggest adding somehow the dates to the trajectories in Figure 1 or adding discussion in the discussion section.

b) The average RH on Mt Tai was low (17%). If indeed aqueous phase processing was a major contributor to the target compounds, RH would have needed to be high during the transport. Can the trajectories tell anything about clouds and/or high RH fields the air masses experienced during transport? Could any precipitation during transport explain an observed decrease in concentrations?

c) The discussion of the concentrations of diacids in the various clusters is not clear. How was the number of 73% of total dicarboxylic acids and related compounds in clusters 2 and 4 determined (p. 7, l. 8)?

3) Mass fraction of total and individual dicarboxylic acids etc a) The authors point out

in the introduction that dicarboxylic acids are 'significant constituents in PM2.5' and '…impact on air quality' (p. 3, l. 4ff). However, later in the text the author quantify that these compounds contribute on average < 3% of OC (p. 7, l. 26), and related compounds even less (< 1%). What is the overall fraction of these compounds in the aerosol, i.e. not only related to organic mass but to total mass? The total (organic + inorganic) mass should be also reported in Table 1. Given these rather small fractions, the text should be reworded accordingly. b) The authors continue using 'significant' for contributions of > 2% within the diacid mass (p. 8, l. 5). I would not call such masses 'significant' given that these species contribute overall < 0.1% tot the total organic mass in the aerosol.

4) VOC measurements a) Only the total VOC mixing ratios are reported, assuming that all of them could be precursors for the target species. Figure 2 might be more meaningful if only a few selected VOCs are shown that have been shown to act as precursors for the identified compounds in Table 1. b) I think it is rather unusual that VOC measurements were performed at University of Irvine but neither in the acknowledgement nor in the author list anyone from this place is listed.

5) Comparison to previous studies How much of diacids and related compounds is expected to be in the size range of PM2.5 to PM10? I.e. is it reasonable to assume that the same scaling factor (0.91) for the total PM2.5/TSP mass can be applied to the diacids and levoglucosan? Are there any measurements (from other locations) that support this assumption?

Minor comments

p. 3, l. 28: 'secondary oxidation' seems redundant

p. 3, l. 15: Are these considered primary sources? In my understanding, combustion of fossil fuel or biomass is an oxidative process and thus the products are secondary.

p. 11, l. 3: What is the meaning of the slope of the correlation?

Table 1: All values should be rounded to significant digits, e.g. 86 +/- 33 instead of 86.2 +/- 33.8

Figure 4: The x-axis is very blurry. I suggest using fewer tick marks.

Technical comments

p. 2, l. 6: 'measure' should be 'measured'

p. 4, l. 22: blank samples

p. 4, l. 24, and remainder of the manuscript: VOC sampling (and VOC samples etc)

p. 6, l. 21: remove 'the trajectories'

p. 6, l. 22: have been given

p. 13, l. 20: 'related' should be 'correlated'

---

## Referee Comment (RC2) · Anonymous Referee #2 · 3 Mar 2018

General Comments:

Zhu et al. discuss trends in concentrations of particle-phase polar oxygenated organic compounds during one month of summer 2014 at Mount Tai. The dataset presented is interesting, particularly in showing daytime versus nighttime measurements, boundary layer height (BLH) estimates, and a broad range of chemical species concentrations with ∼high frequency. The use of principal component analysis (PCA) is also an apt way to summarize potential sources. However, the extent of discussion in the current draft is insufficient for these data and results: each data analysis piece is discussed separately, and cohesion is needed between the BLH estimates, back trajectories, PCA

factors, and concentration trends. I believe the article therefore requires major revisions before final publication in the form of reorganization of the results and discussion, and additional synthesis of the conclusions.

Specific Comments:

There are several pieces of background information that are missing from the introduction. These include brief discussions (with references) of:

- Boundary layer behavior in complex topography;

- Biomass burning emissions and the new regulations mentioned (Pg. 8, line 26); and

- More about general emissions at Mount Tai.

The methods section is lacking key information. Examples of additional information to be included (can go to supplemental material if desired):

- Details about the VOC concentrations: which species do "VOC concentrations" include?

- Discussion of whether the sampling period is representative of Mount Tai during all seasons, years, etc. (concentrations, BLH, and back trajectories)

- The method for calculating limits of detection for the measured chemical species.

- Uncertainties (specify type; e.g., standard deviation) about the measurements of each chemical species reported

- Meteorological conditions and variations during the study

- Frequency of blanks

- A brief synopsis of data used in this article from Zhu et al., 2017

- In the PCA analysis: is the replacement of values below detection limit with have the value a common convention? I am not familiar with this technique, and it seems like it

may bias the measurements low.

-Please report a reference for this if possible, and discuss briefly (this can be in the supporting information).

The results and discussion section should be reorganized to offer a more cohesive analysis of all analytical tools/results. Some specific examples include the following.

- A relationship can be drawn between back trajectory clusters and the chemical concentrations/PCA factors. Do dates of influence of particular source regions align with sources/PCA factors? Do the dominant back trajectory clusters change between the first and second halves of the study, which seem to have different chemical features?

- The authors note that there are relationships between the VOC concentrations and those of the polar organic species measured. Please provide an explanation of what this relationship might be: are the higher concentrations of polar organic species at Mount Tai in 2014 a result of the aging of the measured VOCs? Could they have been directly emitted together as primary aerosol particles? Both? Please support with references. If possible and relevant, please also consider individual VOCs.

- A relationship could be drawn between how BLH estimates might alter the effect of long-range transport (back trajectories) on concentrations. Even if the BLH is only above the sampling location during some sampling times, these could be interesting.

The MGly recovery is estimated to be $\sim$50%. Do the authors expect trends in concentration of MGly, then, to be meaningful? Why is the recovery of Gly expected to be so different?

Daytime/nighttime differences:

- The daytime/nighttime analysis gives a summary of the results, but provides little explanation for the observations. How are these trends informative? Please explain the hypothesis about aqueous photochemical reactions (pg. 10, line 22) more thoroughly and with references. Can the similarities between daytime/nighttime concentrations be

supported by looking at diurnal changes in relative humidity, or contrast between high-elevation/summit and low-elevation/base measurements of any kind at Mount Tai?

- The authors suggest that the strengths of nighttime vs. daytime correlations in Figures 2 and 3 explain daytime/nighttime ratios reported (although, confusingly, these ratios are ∼1 for most species). However, the correlations are not clearly different (daytime/nighttime) in either figure. Please find agreement between the daytime/nighttime ratios, Figures 2 and 3 correlations, and the hypotheses about diurnal variations in concentrations/atmospheric processes.

- Figure 3 includes one outlying point at ∼1800 ng m-3 C2 and ∼35 ðİıJĞg m-3 SO42-; what is the result of removing this point? This looks to me to be driving the daytime/nighttime difference. There is certainly a relationship between these two chemical species, but this may not be easily related to the iron-oxalate hypothesis drawn.

Although the BLH discussion is essential to this analysis, uncertainty in estimating the BLH using a model at a mountaintop should be discussed briefly. In addition, results from the Mount Tai Experiment (Kanaya et al., 2013) showed that their sampling site was above the BLH during many days, and within the residual layer some nights. Please contrast the estimates of these two studies briefly.

Could any of the back trajectories suggest that regional emissions from the previous day or two impacted the measured concentrations? Long range transport is suggested to be dominant throughout the study, but perhaps regional emissions have been transported aloft due to topography and/or convection.

Contrast with other studies:

- The contrast between this summer 2014 study at Mount Tai and others is informative. However, the ratio used for conversion of TSP to PM2.5 likely introduces large uncertainties. Is this ratio relevant for Mount Tai or the North China Plain? For summer? For 2014? Specify briefly, and consider the degree of confidence that the reader can

have in these estimated concentrations, including significant digits of concentrations reported. Please note that composition is size-dependent for aerosol particles.

- Please consider the season, year, and mountainous/urban/rural category of these studies (include study year in the comparison table as well).

Biomass burning discussion:

- The biomass burning discussion is interesting, but incomplete. The authors draw the conclusion that, "...from 2006 to 2014, biomass burning decreased by about 80%." This conclusion cannot be drawn from the estimated concentration of a single species (levoglucosan). Many factors could confound this relationship, such as atmospheric oxidant concentrations, or meteorology during the study. Please rephrase and support with additional observations.

- Please include more information about the "emission hotspots" mentioned on pg. 11, line 27, along with references. Are these the locations of biomass burning events? On a related note, please discuss whether there is any indication that biomass burning events decreased between the first and second halves of the study (satellite data, perhaps). Do trends in concentrations match observations in biomass burning events?

PCA analysis:

- Please be more explicit about the methods and the vocabulary used to describe the results. Specifically, in the methods section, the authors should include not only the information at the beginning of section 3.6, but also whether the data were standardized or mean-centered. Are the "weighting factors" the same as the "factor-loadings"? Please label which values are reported in the table.

- A distinction is made between daytime and nighttime concentrations in the PCA analysis, and slightly different factors are identified. Please provide explanations for differences between all of the daytime and nighttime factors. (In the case of the nighttime factor 4, mixed marine and plastic burning emissions are suggested—please explain

further and cite references.) Please consider agricultural activities as a possible emissions source.

Minor Comments (please change with revisions):

The phrase "dicarboxylic acids and related compounds" is overused in the paper, and must be abbreviated for clarity. Please find an appropriate way to do so. An example might be "polar organic compounds (POCs)".

When reporting values summarizing the campaign data, be clear about whether the value is a mean, etc., in every case.

Throughout the document, please revise for grammar and accuracy of the wording. For example, on pg. 7, line 29, "trends" should be "concentrations".

Please choose a consistent spelling and format for the following terms: "airmass", "daytime" vs. "day", "nighttime" vs. "night", "back trajectory" vs. "back-trajectory".

Pg. 3, line 8: Is this really true that dicarboxylic acids and related compounds are typically studied in TSP rather than PM2.5? Please revisit.

Section 3.2 (and throughout): The discussion of the contributors to "dicarboxylic acids and related compounds" would be much stronger with some context (rather than an empirical grouping of chemicals based on methods). What does this category of chemicals represent in the atmosphere? Could it be representative of water soluble organic carbon? Oxygenated organic species in general? Please support this with references.

Top of pg. 8: Please clarify the definition of each of the percentages reported here. Are these all percentages of the total dicarboxylic acids concentration?

There are several scientific language choices that should be reconsidered: "significant" should be used only when statistical significance is demonstrated; "levels" of chemicals is not precise - please use "concentrations"; "considerable amount" is not precise – please use "substantial concentration", for example.

[Figure]

Where coefficients of determination are discussed, please also report the values within the text.

Please introduce each chemical abbreviation in the article body (e.g., "C2" for oxalic acid, "VOCs" for volatile organic compounds).

Note that phthalic acid and azelaic acid both have primary as well as secondary atmospheric sources.

"Boundary layer height" should be consistently abbreviated to "BLH".

References

Kanaya, Y., Akimoto, H., Wang, Z.-F., Pochanart, P., Kawamura, K., Liu, Y., Li, J., Komazaki, Y., Irie, H., Pan, X.-L., Taketani, F., Yamaji, K., Tanimoto, H., Inomata, S., Kato, S., Suthawaree, J., Okuzawa, K.,, Wang, G., Aggarwal, S. G., Fu, P. Q., Wang, T., Gao, J., Wang, Y., and Zhuang, G.: Overview of the Mount Tai Experiment (MTX2006) in central East China in June 2006: studies of significant regional air pollution, Atmos. Chem. Phys., 13, 8265-8283, 2013.

Zhu, Y., Yang, L., Kawamura, K., Chen, J., Ono, K., Wang, X., Xue, L., and Wang, W.: Contributions and source identification of biogenic and anthropogenic hydrocarbons to 15 secondary organic aerosols at Mt. Tai in 2014, Environ. Pollut., 220, 863-872, 2017.

---

## Author Comment (AC1) · 15 May 2018

Dear Editor and Reviewers,

The authors would like to thank the editor and the reviewers for the constructive and good suggestions to improve our manuscript! We have carefully considered all the review comments and revised the manuscript. Below, we provide responses to the comments, with changes made in the manuscript highlighted in red.

Sincerely

Lingxiao Yang

Ph.D., Professor

Environment Research Institute

Shandong University

Jinan 250100

P. R. China

**Response to Reviewer 1:**

*The authors present data from measurements at Mt Tai focusing on dicarboxylic acids and related compounds. Based on back trajectory and model analysis they conclude that aerosol arriving on Mt Tai has undergone long range transport and has a variety of sources, including anthropogenic emissions and biomass burning. This study is with some new features (back trajectories, WRF model) which, however, have not been really made use of. Overall, it may be an interesting data set, in particular as it is discussed in the context of previous measurements of the same compounds at many different locations and also at Mt Tai in 2006. However, I think the discussion is quite confusing and needs major revision.*

Response: We appreciate the reviewer for the comments and suggestions. We have revised the manuscript accordingly and here address individually the review

comments. WRF model and back trajectories have been studied throughout the draft. Detailed descriptions about aqueous phase oxidation are also presented. Other content cohesions also have been done. A tight and clear manuscript has been obtained. For clarity, the reviewer's comments are listed below in black italics, while our responses and changes in manuscript are shown in blue and red, respectively. Revised table and figure are in the end.

*1. This study is very similar to previous studies from some of the same authors (27% similarity rate), with some new features (back trajectories, WRF model) which, however, have not been really made use of.*

Response: About 15% similarity rate was caused by the name "dicarboxylic acids and related compounds" or the name "dicarboxylic acids, oxocarboxylic acids and α-dicarbonyls". We have changed "dicarboxylic acids and related compounds" and "dicarboxylic acids, oxocarboxylic acids and α-dicarbonyls" to "DCRCs". Moreover, we rephrased the similar expressions sentence by sentence according to the similarity report, please see more details in the revised manuscript.

*2. Day vs night time samples. The authors find that day and night time samples show almost identical concentrations. However, I am not sure that distinguishing day- and night-time-samples is really meaningful here: If a sample was collected at nighttime (i.e. 6 pm - 6 am), it was likely processed during the day(s) before. The same might be true for day time samples that travelled to the sample location for several days. Thus, I am not surprised that samples collected during day and night show very similar composition and loadings.*

Response: DCRCs (see above on this abbreviation) concentrations can be influenced during the day by higher chemical formations due to the occurring photochemistry. On the other hand, DCRCs, such as oxalic acid, can act in the atmosphere as efficient complexing agent forming iron-oxalate complexes which can be effectively photolysis during the day. Thus, the photochemistry may also represent an important sink.

It is expected that DCRCs are secondary products of atmospheric chemistry processes, therefore, the investigation of DCRCs concentration differences during the

day and night conditions has been done. The performed day/night sampling is not ideal and has limitations due to the measured DCRCs form the processing during the day (s) before. Therefore, a sentence addressing this issue has been added to the revised manuscript as follows:

In order to identify the impact of atmospheric chemistry processes on DCRCs, $PM_{2.5}$ samples were collected during the day and night, respectively, from 4 June to 4 July 2014.

(Page 5, Line 7-9)

Choosing 06:00-18:00 and 18:00-06:00 local time as day and night sampling period were due to diurnal variation of UV radiation at the ground of Mt. Tai (Fig. S1). Figure S1 shows an increase from 06:00 when UV radiation begin, followed by a sharp increase and a maximum at 12:00, and a sharp decrease until 18:00. From 19:00 to 23:00 and from 0:00 to 5:00, UV radiation is close to zero.

Moreover, a number of published papers, performing day and night sampling between 06:00-18:00 and 18:00-06:00 local time, showed different diurnal variation, for example Pavuluri et al. (2010), Miyazaki et al. (2009) and Fu et al. (2008). Pavuluri et al. (2010) reported that mostly DCRCs presented clear diurnal trend. Miyazaki et al. (2009) reported that higher concentrations of mostly DCRCs were observed in the night samples. Fu et al. (2008) reported that most of organic compound classes showed higher concentrations in the night samples. We have added descriptions about choosing 06:00-18:00 and 18:00-06:00 local time as sampling time in the revised manuscript as follows:

Considering the diurnal variation of ultraviolet radiation around Mt. Tai (Fig. S1), 06:00-18:00 and 18:00-06:00 local time were selected as the sampling times for day and night, respectively.

(Page 5, Line 9-12)

Reference:

Pavuluri, C.M., Kawamura, K., and Swaminathan, T. Water-soluble organic carbon, dicarboxylic

acids, ketoacids, and a-dicarbonyls in the tropical Indian aerosols. Journal of Geophysical Research, 2010, 115.

Miyazaki, Y., Aggarwal, S.G., Singh, K., Gupta, P.K., Kawamura, K. Dicarboxylic acids and water-soluble organic carbon in aerosols in New Delhi, India, in winter: Characteristics and formation processes. Journal of Geophysical Research, 2009, 114.

Fu, P.Q., Kawamura, K., Okuzawa, K., Aggarwal, S.G., Wang, G.H., Kanaya, K., Wang, Z.F. Organic molecular compositions and temporal variations of summertime mountain aerosols over Mt. Tai, North China Plain, 2008. Journal of Geophysical Research, 2008, 113.

*Unless I misunderstood the sampling and nomenclature of day/night samples, I suggest removing the discussion of day- versus night-samples. That way, hypotheses such as on night time oxidation (p. 2, l. 12) or less effective loss during night (p. 10, l. 24) could be removed as they do not seem supported.*

Response: Our results showed the day concentrations of DCRCs were similar with their night concentrations. The day-night concentration ratios of 79% DCRCs individual species ranged between 0.9 and 1.1. However, 2006 results showed day concentrations of DCRCs were 2-3 times higher than those in the night (Wang et al., 2009). In order to accurately identify major control factor of DCRCs at Mt. Tai, such as boundary layer heights, long-range transport, ground pollutants transport and aqueous phase oxidation, day/night variations in this study were necessary to discuss.

Reference:

Wang, G.H., Kawamura, K., Umemoto, N., Xie, M.J., Hu, S.H., Wang, Z.F. Water-soluble organic compounds in PM2.5 and size-segregated aerosols over Mount Tai in North China Plain. Journal of Geophysical Research, 2009, 114, D19208.

*3. Trajectories a) In Figure 1, the authors show 72 h-back trajectories of air masses arriving at Mt Tai. These trajectories are briefly discussed in Section 3.1. However, the authors do not link their later discussion to these trajectories. For example, there seems to be change in conditions (meteorology, emissions, air mass?) after the first half of the sampling period that leads to lower diacid loadings. Could that be linked to different trajectories? I suggest adding somehow the dates to the trajectories in*

*Figure 1 or adding discussion in the discussion section.*

Response: 4-19 June was the first half sampling period, and 20 June-04 July was the second half sampling period. Sampling dates in different clusters as follows:

Cluster 1: 4-5 June, 29 June

Cluster 2: 6 June, 11-13 June, 22-24 June, 28 June, 4 July

Cluster 3: 7-8 June

Cluster 4: 9-10 June, 14-21 June, 25-27 June, 30 June, 1-3 July.

We have added sampling dates in 3.1 discussion part as follows:

Sampling dates in cluster 2 included 6 June, 11-13 June, 22-24 June, 28 June and 4 July, while 9-10 June, 14-21 June, 25-27 June, 30 June and 1-3 July belonged to cluster 4.

(Page 8, Line 23-25)

Trajectories on 4-5 June and 29 June were grouped into cluster 1, while trajectories on 7-8 June were grouped into cluster 3.

(Page 9, Line 1-2)

Impact of air mass and meteorology on DCRCs concentrations in the first and second half sampling periods have been discussed and added in the revised manuscript as follows:

From the trajectory analysis, we can see that during the first and second half of the sampling periods, 4 and 5 days, respectively, belonged to cluster 2. In addition, 8 and 9 days belonged to cluster 4, respectively. Therefore, the dominant air masses in the first and second half of the sampling periods were similar, and thus had a low impact on DCRC concentrations in the two periods. Figure S2 shows meteorological data in the different backward trajectory clusters during the sampling period at Mt. Tai. We can see that pressure, temperature and RH didn't change much in clusters 2 and 4. Moreover, as shown in Fig. 1, meteorological data at Mt. Tai site also didn't change much between the first and second half of the sampling periods. Therefore, the quite stable meteorological conditions may have had a low impact on the DCRC concentrations between the first and second half of the sampling periods.

Higher DCRCs concentrations in the first half sampling period were most likely caused by biomass burning. Dicarboxylic acids and $K^+$ showed strong correlation in the first half, while in the second half, dicarboxylic acids and $K^+$ exhibited no correlation (Fig. 8). Moreover, as shown in Fig. 9, straw burning hotspots mainly distributed in the first half sampling period, while in the second half sampling period, hotspots almost disappeared. Detailed descriptions about biomass burning have been showed in the revised manuscript as follows:

Dicarboxylic acids and $K^+$ exhibited a strong correlation during the first half, while during the second half, dicarboxylic acids and $K^+$ exhibited no correlation (Fig. 8). The peaks of dicarboxylic acids and $K^+$ appeared almost simultaneously (Fig. 7). It is also clear that when the $K^+$ concentration increased, dicarboxylic acids correspondingly increased during the first half (Fig. 8). These results imply that biomass burning was an important contributor to DCRCs during the first half of the measurement period. Moreover, according to reports by weather satellites of the Ministry of Environment Protection of the People's Republic of China (http://www.zhb.gov.cn/), straw burning hotspots in air masses that passed over key areas (Anhui, Hebei and Shandong province) were mainly distributed in the first half of the sampling period (Fig. 9). This result further supports that biomass burning was more important in the first half of the sampling period.

*b) The average RH on Mt Tai was low (17%). If indeed aqueous phase processing was a major contributor to the target compounds, RH would have needed to be high during the transport. Can the trajectories tell anything about clouds and/or high RH fields the air masses experienced during transport? Could any precipitation during transport explain an observed decrease in concentrations?*

Response: Please note in 2.1 part, we reported that "During sampling period, average values of temperatures, relative humidity (RH) and winds were 17 °C, 87% and 2.1 m/s." Average RH values at Mt. Tai were 87%, not 17%.

As shown in Fig. 1, average RH values during the sampling period at Mt. Tai were 87%, up to 100%. Moreover, as mentioned above, average RH values were 67% and 72% in clusters 2 and 4 (they were dominant back-trajectory clusters) (Fig. S2), respectively. However, due to the coarse resolution of HYSPLIT, it was difficult to judge whether clouds occurred. Therefore, MODIS satellite pictures were investigated, and the results showed sometimes clouds occurred in the region of Mt. Tai and in the areas of trajectories passed over during the sampling period. But MODIS satellite pictures limited information about the cloud base and top heights, thus it cannot exactly explore whether there were clouds in the height of the trajectories.

As shown in Fig. S2, we can see that precipitation during the back-trajectory clusters was low, the maximum of precipitation was just 0.11 mm, almost zero. During the campaign at Mt. Tai, minor rain events occurred on 15, 16 June and 3 July, major rain events occurred on 24 June and 4 July. Therefore, precipitation was also likely not the reason for the decrease of DCRCs concentrations in the second half sampling period.

We have added some descriptions about RH in 3.4 discussion part as follows:

Average RH values during the sampling period at Mt. Tai were 87%, up to 100% (Fig. 1), and higher on average during the night (Fig. 5). In addition, average RH values along the dominant back-trajectory clusters (clusters 2 and 4) were about 70% (Fig. S2). However, due to the coarse resolution of HYSPLIT, it was difficult to judge whether clouds occurred. Therefore, MODIS satellite pictures were investigated, and the results showed that clouds sometimes occurred in the region of Mt. Tai and in the areas that the trajectories passed over during the sampling period. But MODIS satellite pictures have limited information about the cloud base and top heights, and thus it cannot exactly explore whether there were clouds at the height of the trajectories.

(Page 12, Line 27-Page 13, Line 7)

We have added descriptions about precipitation as follows:

Minor rain events occurred on 15, 16 June and 3 July, and major rain events occurred

on 24 June and 4 July. The sample collection was ended just before the major rain.

(Page 5, Line 2-3)

*c) The discussion of the concentrations of diacids in the various clusters is not clear. How was the number of 73% of total dicarboxylic acids and related compounds in clusters 2 and 4 determined (p. 7, l. 8)?*

Response: We have revised the sentence as follows:

The sum of DCRC concentrations in clusters 2 and 4 contributed 73% of the total concentration of DCRCs during the sampling period.

(Page 8, Line 21-22)

*4. Mass fraction of total and individual dicarboxylic acids etc a)-1 The authors point out in the introduction that dicarboxylic acids are 'significant constituents in PM2.5' and ': : :impact on air quality' (p. 3, l. 4ff). However, later in the text the author quantify that these compounds contribute on average < 3% of OC (p. 7, l. 26), and related compounds even less (< 1%). What is the overall fraction of these compounds in the aerosol, i.e. not only related to organic mass but to total mass?*

Response: DCRCs are not only oxygenated organic compounds but water-soluble components of SOA (Kawamura and Sakaguchi, 1999; Kawamura and Yasui, 2005; Pavuluri et al., 2010). Until now only a small fraction of water-soluble and oxygenated SOA can be identified in a compound specific manner. Zhu et al. (2017) reported that only 18.1-49.1% of SOA can be detected, a large fraction can't be identified. Therefore, DCRCs are important constituents in the water soluble and oxygenated matter of $PM_{2.5}$. We have changed related descriptions in the revised manuscript as follows:

Dicarboxylic acids and related compounds (oxocarboxylic acids and α-dicarbonyls) (DCRCs) are important constituents in $PM_{2.5}$, and mainly produced by secondary processes (Kawamura and Yasui, 2005; Pavuluri et al., 2010a). Due to their high water solubility, DCRCs contribute to the water soluble organic fraction of $PM_{2.5}$, which can have an impact on air quality (van Pinxteren et al., 2009; Kawamura and Bikkina, 2016; Kundu et al., 2010b).

(Page 3, Line 2-7)

PM$_{2.5}$ mass concentration at Mt. Tai during the campaign was 98.2 ± 29.2, 98.6 ± 25.3 μg m$^{-3}$ in the day and night, respectively. DCRCs concentration contributed 1.2% and 1.1% to PM$_{2.5}$ in the day and night, respectively. The DCRCs-C at Mt. Tai in 2014 accounted for 3.24% and 3.20% of OC in the day and night, respectively. We have added these results in the revised manuscript as follows:

PM$_{2.5}$ mass concentration at Mt. Tai during the campaign was 98.2 ± 29.2 and 98.6 ± 25.3 μg m$^{-3}$ during the day and night, respectively. DCRC total concentration contributed about 1.2% and 1.1% to PM$_{2.5}$ in the day and night, respectively. In addition, the DCRCs-C accounted for 3.3% and 3.2% of OC in the day and night, respectively.

(Page 9, Line 25-29)

Response: changed.

[revised manuscript text omitted]

---

## Author Comment (AC2) · 15 May 2018

Dear Editor and Reviewers,

The authors would like to thank the editor and the reviewers for the constructive and good suggestions to improve our manuscript! We have carefully considered all the review comments and revised the manuscript. Below, we provide responses to the comments, with changes made in the manuscript highlighted in red.

Sincerely

Lingxiao Yang

Ph.D., Professor

Environment Research Institute

Shandong University

Jinan 250100

P. R. China

**Response to Reviewer 2:**

*Zhu et al. discuss trends in concentrations of particle-phase polar oxygenated organic compounds during one month of summer 2014 at Mount Tai. The dataset presented is interesting, particularly in showing daytime versus nighttime measurements, boundary layer height (BLH) estimates, and a broad range of chemical species concentrations with ~high frequency. The use of principal component analysis (PCA) is also an apt way to summarize potential sources. However, the extent of discussion in the current draft is insufficient for these data and results: each data analysis piece is discussed separately, and cohesionis needed between the BLH estimates, back trajectories, PCA factors, and concentration trends. I believe the article therefore requires major revisions before final publication in the form of reorganization of the results and discussion, and additional synthesis of the conclusions.*

Response: We thank the reviewer for the helpful comments. Below we address the comments and have revised the manuscript accordingly. For clarity, the reviewer's comments are listed below in black italics, whilst our responses and changes in manuscript are shown in blue and red, respectively. Revised table and figure are in the end.

*1. There are several pieces of background information that are missing from the introduction. These include brief discussions (with references) of:*

*- Boundary layer behavior in complex topography;*

*- Biomass burning emissions and the new regulations mentioned (Pg. 8, line 26);*

*and - More about general emissions at Mount Tai.*

Response: We have added background information about boundary layer behavior, biomass burning emissions and the new regulations mentioned, and emissions at Mount Tai as follows:

Moreover, it should be noted that mountain areas, with parallel ridges or isolated ridges and peaks, are different from the plain in terms of geometric structures. This has implications for modifying the ambient air flow by this complex terrain, which leads to complexity of the mountain boundary layer structure (Smith et al., 2002). Naturally, the boundary layer structure plays important roles in the transport and dispersal of atmospheric pollutants during long-range transport (Garratt, 1994).

(Page 4, Line 1-7)

In addition, the NCP is one of the most productive agricultural regions in China, and agricultural waste burning occurs frequently during the harvest seasons. Although some management strategies have been implemented by the Chinese government, such as lawful punishment or punishment by a fine, biomass burning still occurs during the harvest seasons (Zhu et al., 2017).

(Page 3, Line 15-20)

There are many tourists at Mt. Tai in summer, so there are some local emissions from small restaurants and temples (Gao et al., 2005). Furthermore, 80% of Mt. Tai is covered by vegetation (mostly bushes).

(Page 3, Line 28-Page 4, Line 1)

*2. The methods section is lacking key information. Examples of additional information to be included (can go to supplemental material if desired):*

*- Details about the VOC concentrations: which species do "VOC concentrations" include?*

Response: Detailed descriptions about VOCs concentrations at Mt. Tai in 04 June-04 July 2014 have been given in our previous study-Zhu et al. (2017). Therefore, in this study, we briefly described the measured VOCs.

Several studies have demonstrated that, e.g., ethyne, ethene, isoprene, α-pinene, β-pinene, toluene, m/p-xylene, o-xylene represents potential DCRCs precursors (Warneck, 2003; Ervens et al., 2004; Bikkina et al., 2014; Tilgner and Herrmann, 2010). Therefore, in this study, we selected ethyne, ethene, isoprene, α-pinene, β-pinene, toluene, m/p-xylene, o-xylene as DCRCs precursors. Then, we made a scatter plot using total concentration of selected DCRCs precursors and DCRCs total concentration. The result was presented in Fig. 4.

We also added descriptions about DCRCs precursors as follows:

As shown in Fig. 4, DCRC concentrations exhibited weak and moderate correlation with total concentration of selected DCRC precursors in the day and night, respectively, where selected DCRCs precursors included ethyne, ethene, isoprene, α-pinene, β-pinene, toluene, m/p-xylene and o-xylene (Warneck, 2003; Ervens et al., 2004; Bikkina et al., 2014; Tilgner and Herrmann, 2010).

(Page 12, Line 18-22)

Reference:

Zhu, Y., Yang, L., Kawamura, K., Chen, J., Ono, K., Wang, X., Xue, L., and Wang, W.: Contributions and source identification of biogenic and anthropogenic hydrocarbons to secondary organic aerosols at Mt. Tai in 2014, Environ. Pollut., 220, 863-872,2017.

Warneck, P.: In-cloud chemistry opens pathway to the formation of oxalic acid in the marine atmosphere, Atmos. Environ., 37, 2423–2427, 2003.

Ervens, B., Feingold, G., Frost, G, J., and Kreidenweis, S. M.: A modeling study of aqueous

production of dicarboxylic acids: 1. Chemical pathways and speciated organic mass production, J. Geophys. Res., 109, D15205, doi:10.1029/2003JD004387, 2004.

Bikkina, S., Kawamura, K., Miyazaki, Y., Fu, P., 2014. High abundances of oxalic, azelaic, and glyoxylic acids andmethylglyoxal in the open oceanwith high biological activity: implication for secondary OA formation from isoprene. Geophys. Res. Lett. 41, 3649–3657. http://dx.doi.org/10.1002/2014GL059913.

Tilgner, A., and Herrmann, H.: Radical-driven carbonyl-to-acid conversion and acid degradation in tropospheric aqueous systems studied by CAPRAM, Atmos. Environ., 44, 5415-5422, 2010.

*- Discussion of whether the sampling period is representative of Mount Tai during all seasons, years, etc. (concentrations, BLH, and back trajectories).*

Response: In this study, the results were only representative of Mt. Tai in 04 June-04 July 2014. According to straw burning hotspots number from May to November 2014 reported by weather satellites of the Ministry of Environment Protection of the People's Republic of China (Fig. R1), we can see that June can represent the wheat burning season in 2014. Therefore, we changed title of the draft to "Molecular distributions of dicarboxylic acids, oxocarboxylic acids and α-dicarbonyls in $PM_{2.5}$ collected at the top of Mount Tai, in North China during wheat burning season 2014".

[Figure]

Fig. R1. Straw burning hotspots number in China from May to November 2014.

*- The method for calculating limits of detection for the measured chemical species.*

Response: Detection limits of the measured chemical species were calculated on the basis of minimum areas, which were set at usually 500 counts in Shimadzu CR7 integrator.

We have added method for calculating detection limits as follows:

Detection limits of the measured chemical species were 0.05 to 0.1 ng m$^{-3}$, which were calculated on the basis of minimum areas.

(Page 6, Line 25-27)

*-Uncertainties (specify type; e.g., standard deviation) about the measurements of each chemical species reported*

Response: Standard deviation is used to quantify the amount of variation of a set of data values. Therefore, we use standard deviation to present measured concentration of each chemical species. The standard deviation of measured DCRCs individual species has been provided in the revised manuscript. Please see Table 1.

*- Meteorological conditions and variations during the study*

Response: We have added descriptions about meteorological conditions and variations during the sampling period and shown in Fig. 1. Moreover, meteorological conditions have been added in the revised manuscript as follows:

The meteorological data during the sampling period are summarized in Fig. 1. The ambient temperatures covered a range of 10-25 ℃ with an average of 17 ℃. Relative humidity (RH) varied between 58 to 100% with an average of 87%. Winds generally came from the northwest, and wind speeds ranged from 1-7 m s$^{-1}$. Weather conditions during the campaign were mostly cloudy and occasionally foggy. Minor rain events occurred on 15, 16 June and 3 July, and major rain events occurred on 24 June and 4 July. The sample collection was ended just before the major rain.

(Page 4, Line 24-Page 5, Line 3)

*- Frequency of blanks*

Response: We have added the frequency of blanks as follows:

Blank samples were collected between 06:00-18:00 and 18:00-06:00 local time from 5-7 July 2014, and their sampling manner was similar to the real samples, but without pumping.

(Page 5, Line 12-15)

*- A brief synopsis of data used in this article from Zhu et al., 2017*

Response: We have added a brief synopsis of data used in this study from Zhu et al. (2017) as follows:

In our previous publication (Zhu et al., 2017), we described the meteorological conditions, sampling site, $PM_{2.5}$ sampling, VOCs sampling and analysis from 04 June to 04 July 2014 at Mt. Tai. Therefore, in this study we describe these experimental methods only briefly.

(Page 4, Line 16-19)

*- In the PCA analysis: is the replacement of values below detection limit with have the value a common convention? I am not familiar with this technique, and it seems like it may bias the measurements low.*

Response: According to Wold et al. (1987), when compound concentration was below the detection limit, the data was replaced by a value half of the corresponding detection limit. The method has been used in van Pinxteren et al. (2014).

Reference:

van Pinxteren, D., Neusüß, C., and Herrmann, H.: On the abundance and source contributions of dicarboxylic acids in size-resolved aerosol particles at continental sites in central Europe, Atmos. Chem. Phys., 14, 3913-3928, 2014.

Wold, S., Esbensen, K., Geladi, P.: Principal component analysis, Chemometr. Intell. Lab., 2, 37-52, 1987.

*-Please report a reference for this if possible, and discuss briefly (this can be in the supporting information).*

Response: We have added reference in the revised manuscript as follows:

If the compound concentration was below the detection limit, the data were replaced by a value half of the corresponding detection limit (Wold et al., 1987).

(Page 15, Line 28-Page 16, Line 1)

*3. The results and discussion section should be reorganized to offer a more cohesive analysis of all analytical tools/results. Some specific examples include the following.*

*- A relationship can be drawn between back trajectory clusters and the chemical concentrations/PCA factors. Do dates of influence of particular source regions align with sources/PCA factors? Do the dominant back trajectory clusters change between the first and second halves of the study, which seem to have different chemical features?*

*(1) A relationship can be drawn between back trajectory clusters and the chemical concentrations.*

Response: We have connected DCRCs concentrations with back-trajectory clusters as follows:

According to back-trajectories and classification results of DCRCs in different back-trajectory clusters in Fig. 2, we can see that DCRCs concentrations were mostly higher in air masses that originated from north of Mt. Tai (northern Hebei province) (31%) and south of Mt. Tai (northern Anhui province) (48%), but lower in air masses derived from the ocean (11%) and Siberia (10%).

(Page 9, Line 20-25)

*(2) A relationship can be drawn between back trajectory clusters and the PCA factors.*

Response: We have added back-trajectory parameters (mean trajectory length, solar flux along trajectory, mixing depth along trajectory) in PCA analysis as follows:

The negative loading of mean trajectory length and mixing depth along the trajectory to PC1 suggested high residence times of trajectories above continental areas.

(Page 16, Line 9-11)

The low loading of solar flux in PC2 indicated that secondary sources were not primarily driven by radiation.

(Page 16, Line 13-14)

The negative loading of mean trajectory length and mixing depth along the trajectory in PC1, PC2 and PC3 indicates long residence times above the continental areas.

(Page 17, Line 3-5)

*(3) Do dates of influence of particular source regions align with sources/PCA factors?*

Response: Using PCA method, DCRCs sources in the first and second half sampling periods were identified, respectively. As shown in Table R1, sources in principal components (PCs) in the two periods were as follows:

In the first half sampling period:

PC 1: anthropogenic activities (such as fossil fuel and biomass burning) followed by photochemical aging

PC 2: secondary sources

PC 3: fuel combustion, solid wastes/plastic polymers burning

PC 4: photooxidation of unsaturated fatty acids emitted from the sea surface together

with sea salt particles

In the second half sampling period:

PC 1: anthropogenic activities (such as fossil fuel combustion) followed by photochemical aging

PC 2: secondary sources

PC 3: fuel combustion, solid wastes/plastic polymers burning

PC 4: photooxidation of unsaturated fatty acids emitted from the sea surface together with sea salt particles

We can see that only biomass burning was different in the two periods, other sources were the same in the two periods. Biomass burning was important in the first half sampling period, while in the second half sampling period, biomass burning had no impact. 3.5 discussion part has presented detailed descriptions about the importance of biomass burning during the first half sampling period.

Table R1. PCA analysis results for DCRCs in the first and second half sampling periods, respectively.

| Compounds | The first half sampling period | | | | The second half sampling period | | | |
|---|---|---|---|---|---|---|---|---|
| $C_2$ | **0.765** | 0.339 | 0.409 | | **0.734** | 0.367 | 0.411 | |
| $C_3$ | **0.879** | 0.318 | | | **0.882** | 0.302 | | |
| $C_4$ | **0.725** | 0.402 | 0.512 | | **0.697** | 0.382 | 0.523 | |
| $C_5$ | **0.798** | | 0.590 | | 0.484 | | **0.605** | |
| $C_6$ | 0.570 | | | | 0.371 | | | |
| $C_9$ | | | | 0.772 | | | | 0.792 |
| $iC_5$ | **0.683** | | 0.619 | | **0.637** | | 0.652 | |
| M | | | **0.824** | | | | **0.847** | |
| F | 0.592 | 0.207 | **0.607** | | **0.627** | | **0.613** | |
| $hC_4$ | **0.609** | 0.197 | | | 0.502 | | | |
| Ph | **0.616** | | 0.398 | | **0.639** | | 0.351 | |

| | | | | | | | | |
|---|---|---|---|---|---|---|---|---|
| tPh | | | **0.626** | | | | **0.661** | |
| kC$_3$ | **0.670** | | | | **0.606** | | | |
| Pyr | **0.768** | 0.332 | | | **0.731** | 0.393 | | |
| ωC$_2$ | **0.812** | 0.397 | | | **0.797** | 0.342 | | |
| ωC$_4$ | **0.809** | | | | **0.693** | | | |
| Gly | **0.813** | 0.396 | | | **0.761** | 0.304 | | |
| MGly | **0.618** | 0.429 | | | 0.582 | 0.471 | | |
| OC | **0.728** | | | 0.470 | **0.717** | | | 0.528 |
| EC | 0.428 | 0.186 | **0.679** | | 0.457 | | **0.702** | |
| Na$^+$ | | 0.339 | **0.832** | | | 0.359 | | **0.857** |
| NH$_4^+$ | 0.365 | **0.827** | | | 0.380 | **0.892** | | |
| K$^+$ | **0.709** | 0.253 | | | | | | |
| NO$_3^-$ | 0.239 | **0.792** | 0.397 | | 0.253 | **0.778** | 0.278 | |
| SO$_4^{2-}$ | 0.272 | **0.819** | | | 0.277 | **0.719** | | |
| Mean trajectory length | **-0.645** | -0.589 | -0.535 | | **-0.683** | -0.602 | -0.589 | |
| Solar flux along trajectory | 0.328 | 0.389 | | 0.272 | 0.356 | 0.210 | | |
| Mixing depth along trajectory | -0.478 | -0.492 | -0.361 | | -0.519 | -0.418 | -0.432 | |
| Variance (%) | 68% | 10% | 9% | 6% | 57% | 18% | 11% | 7% |

*(1) Do the dominant back trajectory clusters change between the first and second halves of the study, which seem to have different chemical features?*

Response: 4-19 June was the first half sampling period, and 20 June-04 July was the second half sampling period. Sampling dates in different clusters as follows:

Cluster 1: 4-5 June, 29 June

Cluster 2: 6 June, 11-13 June, 22-24 June, 28 June, 4 July

Cluster 3: 7-8 June

Cluster 4: 9-10 June, 14-21 June, 25-27 June, 30 June, 1-3 July.

We have added sampling dates in 3.1 discussion part as follows:

Sampling dates in cluster 2 included 6 June, 11-13 June, 22-24 June, 28 June and 4 July, while 9-10 June, 14-21 June, 25-27 June, 30 June and 1-3 July belonged to cluster 4.

(Page 8, Line 23-25)

Trajectories on 4-5 June and 29 June were grouped into cluster 1, while trajectories on 7-8 June were grouped into cluster 3.

(Page 9, Line 1-2)

Impact of air mass and meteorology on DCRCs concentrations in the first and second half sampling periods have been discussed and added in the revised manuscript as follows:

From the trajectory analysis, we can see that during the first and second half of the sampling periods, 4 and 5 days, respectively, belonged to cluster 2. In addition, 8 and 9 days belonged to cluster 4, respectively. Therefore, the dominant air masses in the first and second half of the sampling periods were similar, and thus had a low impact on DCRC concentrations in the two periods. Figure S2 shows meteorological data in the different backward trajectory clusters during the sampling period at Mt. Tai. We can see that pressure, temperature and RH didn't change much in clusters 2 and 4. Moreover, as shown in Fig. 1, meteorological data at Mt. Tai site also didn't change much between the first and second half of the sampling periods. Therefore, the quite stable meteorological conditions may have had a low impact on the DCRC concentrations between the first and second half of the sampling periods.

(Page 13, Line 29-Page 14, Line 10)

Higher DCRCs concentrations in the first half sampling period were most likely caused by biomass burning. Dicarboxylic acids and $K^+$ showed strong correlation in the first half, while in the second half, dicarboxylic acids and $K^+$ exhibited no correlation (Fig. 8). Moreover, as shown in Fig. 9, straw burning hotspots mainly distributed in the first half sampling period, while in the second half sampling period, hotspots almost disappeared. Detailed descriptions about biomass burning have been showed in the revised manuscript as follows:

Dicarboxylic acids and $K^+$ exhibited a strong correlation during the first half, while during the second half, dicarboxylic acids and $K^+$ exhibited no correlation (Fig. 8). The peaks of dicarboxylic acids and $K^+$ appeared almost simultaneously (Fig. 7). It is also clear that when the $K^+$ concentration increased, dicarboxylic acids correspondingly increased during the first half (Fig. 8). These results imply that biomass burning was an important contributor to DCRCs during the first half of the measurement period. Moreover, according to reports by weather satellites of the Ministry of Environment Protection of the People's Republic of China (http://www.zhb.gov.cn/), straw burning hotspots in air masses that passed over key areas (Anhui, Hebei and Shandong province) were mainly distributed in the first half of the sampling period (Fig. 9). This result further supports that biomass burning was more important in the first half of the sampling period.

(Page 14, Line 11-22)

- *The authors note that there are relationships between the VOC concentrations and those of the polar organic species measured. Please provide an explanation of what this relationship might be: are the higher concentrations of polar organic species at Mount Tai in 2014 a result of the aging of the measured VOCs? Could they have been directly emitted together as primary aerosol particles? Both? Please support with references. If possible and relevant, please also consider individual VOCs.*

Response: When aerosols are aged, $C_4$ will be oxidized to $C_3$. However, in this study, $C_4$ concentrations were much higher than $C_3$. Therefore, it was possible that polar organic compounds together with VOCs originated from primary sources.

[Figure]

**Fig. R2.** Scatter plot of the day and night concentration between ethene/toluene and oxalic acid ($C_2$).

As shown in Fig. R2, ethene or toluene presented correlation with oxalic acid ($C_2$), but not strong. When ethene or toluene concentration increased, $C_2$ concentration also increased. These results suggested the higher concentrations of polar organic species at Mt. Tai in 2014 were likely caused by the aging of the measured VOCs.

*- A relationship could be drawn between how BLH estimates might alter the effect of long-range transport (back trajectories) on concentrations. Even if the BLH is only above the sampling location during some sampling times, these could be interesting.*

Response: We have made a graph about relationship between BLHs and total concentration of DCRCs and shown in Fig. 3.

We have added descriptions about relationship between BLHs and total concentration of DCRCs as follows:

The BLHs were higher during the day, peaking near noontime. The boundary layer occasionally extended high enough during the day to approach the sampling site (Fig. 3). However, the maximum BLH was only ~ 600 m during the night, which was much lower than the sampling site height. As shown in Fig. 3, the total concentration of DCRCs increased when BLHs were higher than the site elevation. These results suggest that mountain/valley breezes may bring ground-level pollutants to the summit of Mt. Tai during the day when the BLHs were above the sampling site height.

*4. The MGly recovery is estimated to be~50%. Do the authors expect trends in concentration of MGly, then, to be meaningful? Why is the recovery of Gly expected to be so different?*

Response: MGly recovery was ca. 50% using authentic standard. The reproducibility of MGly is good enough (analytical errors < 30%) to discuss the changes in the concentrations and trends.

*5. Daytime/nighttime differences:*

*- The daytime/nighttime analysis gives a summary of the results, but provides little explanation for the observations. How are these trends informative? Please explain the hypothesis about aqueous photochemical reactions (pg. 10, line 22) more thoroughly and with references. Can the similarities between daytime/nighttime concentrations be supported by looking at diurnal changes in relative humidity, or contrast between high elevation/summit and low-elevation/base measurements of any kind at Mount Tai?*

Response: Average diurnal change of RH was presented in Fig. 5. And we have added description about average diurnal variation of RH as follows:

Average RH values during the sampling period at Mt. Tai were 87%, up to 100% (Fig. 1), and higher on average during the night (Fig. 5). In addition, average RH values along the dominant back-trajectory clusters (clusters 2 and 4) were about 70% (Fig. S2). However, due to the coarse resolution of HYSPLIT, it was difficult to judge whether clouds occurred. Therefore, MODIS satellite pictures were investigated, and the results showed that clouds sometimes occurred in the region of Mt. Tai and in the areas that the trajectories passed over during the sampling period. But MODIS satellite pictures have limited information about the cloud base and top heights, and thus it cannot exactly explore whether there were clouds at the height of the trajectories.

(Page 12, Line 27-Page 13, Line 7)

During the campaign (04 June-04 July 2014) at Mt. Tai, measured ozone ($O_3$) concentration was $85 \pm 21$ ppb. However, at the same period (04 June-04 July 2014),

O$_3$ concentration at the foot of Mt. Tai was 123 ± 18 ppb (https://www.zq12369.com/environment.php?city). The large difference of O$_3$ concentration between high elevation and low elevation measurements suggested that anthropogenic sources in the valleys had low impact on pollutant concentrations at the top of Mt. Tai. As a consequence of the low BLH and mixing layer height, air masses in the low and high altitude might be separated.

*-The authors suggest that the strengths of nighttime vs. daytime correlations in Figures 2 and 3 explain daytime/nighttime ratios reported (although, confusingly, these ratios are ∼1 for most species). However, the correlations are not clearly different (daytime/nighttime) in either figure. Please find agreement between the daytime/nighttime ratios, Figures 2 and 3 correlations, and the hypotheses about diurnal variations in concentrations/atmospheric processes.*

Response: We have changed Fig. 2, and now shown in Fig. 4. Ethyne, ethene, isoprene, α-pinene, β-pinene, toluene, m/p-xylene and o-xylene were used as DCRCs precursors-VOCs, which were pointed out in published papers as sources of dicarboxylic acids (Warneck, 2003; Ervens et al., 2004; Bikkina et al., 2014; Tilgner and Herrmann, 2010). Then we made scatter plot using total concentration of selected DCRCs precursors and DCRCs concentration. The result was presented in Fig. 4. The moderate correlations suggested that the difference in the day and night measurements was possibly dependent on the amount of precursor emissions.

We added descriptions about DCRCs precursors as follows:

As shown in Fig. 4, DCRC concentrations exhibited weak and moderate correlation with total concentration of selected DCRC precursors in the day and night, respectively, where selected DCRCs precursors included ethyne, ethene, isoprene, α-pinene, β-pinene, toluene, m/p-xylene and o-xylene (Warneck, 2003; Ervens et al., 2004; Bikkina et al., 2014; Tilgner and Herrmann, 2010).

(Page 12, Line 18-22)

We have changed Fig. 3 by deleting outlying point at ∼1800 ng m$^{-3}$ C$_2$ and 35 μg

$m^{-3}$ $SO_4^{2-}$, and now shown in Fig. 6. Higher correlation at the night suggested more important of aqueous phase oxidations at the night.

*- Please include more information about the "emission hotspots" mentioned on pg. 11, line 27, along with references. Are these the locations of biomass burning events? On a related note, please discuss whether there is any indication that biomass burning events decreased between the first and second halves of the study (satellite data, perhaps). Do trends in concentrations match observations in biomass burning events?*

Response: We have shown straw burning hotspots in Fig. 9 and added more information about emission hotspots as follows:

Moreover, according to reports by weather satellites of the Ministry of Environment Protection of the People's Republic of China (http://www.zhb.gov.cn/), straw burning hotspots in air masses that passed over key areas (Anhui, Hebei and Shandong province) were mainly distributed in the first half of the sampling period (Fig. 9). This result further supports that biomass burning was more important in the first half of the sampling period.

(Page 14, Line 17-22)

*10. PCA analysis:*

*- Please be more explicit about the methods and the vocabulary used to describe the results. Specifically, in the methods section, the authors should include not only the information at the beginning of section 3.6, but also whether the data were standardized or mean-centered. Are the "weighting factors" the same as the "factor-loadings"? Please label which values are reported in the table.*

*(1) Specifically, in the methods section, the authors should include not only the information at the beginning of section 3.6, but also whether the data were standardized or mean-centered.*

Response: We have added more descriptions about data standardization and PCA method in the methods section as follows:

**2.6 The principal component analysis (PCA) method**

Principal component analysis (PCA) is a multivariate analytical tool. It starts with a great many correlated variables and attempts to find a smaller number of independent factors, which can explain the variance in data. Here, the compound concentrations should firstly be transformed into standardized form using the following formula:

$$Z_{ij} = \frac{C_{ij} - \overline{C_j}}{\sigma_j}$$

where $i = 1, \cdots, n$ sample; $j = 1, \cdots, m$ compound; $C_{ij}$ is the concentration of compound j in sample i; and $\overline{C_j}$ and $\sigma_j$ are the arithmetic mean concentration and the standard deviation for compound j, respectively. The derived variables are linear combinations of original variables (Callén et al., 2009). In order to better identify the influence of the original variables, varimax rotation is used to obtain the rotated factor loadings that reflect the contribution of each variable to its principal component (PC) (Almeida et al. 2005; Viana et al., 2006). Factor loadings reveal how much a variable contributes to the corresponding PC and how well a variable differs from others. Only factors with eigenvalues greater than 1 are extracted based on Kaiser-Meyer-Olkin (KMO) and the Bartlett's test of sphericity.

(Page 7, Line 22-Page 8, Line 11)

*(2) Are the "weighting factors" the same as the "factor-loadings"?*

Response: The "weighting factors" is the same as the "factor-loadings". In this study, only the term "factor-loadings" is used. We have changed "weighting factors" to "factor-loadings".

*(3) Please label which values are reported in the table.*

Response: We have added descriptions about values reported in the table as follows:

In this study, PCA was employed to identify the DCRC sources in $PM_{2.5}$. Concentrations of $C_2$, $C_3$, $C_4$, $C_5$, $C_6$, $C_9$, $iC_5$, $hC_4$, M, F, Ph, tPh, $kC_3$, Pyr, $\omega C_2$, $\omega C_4$,

Gly, MGly (compound abbreviation in Table 1), OC, EC, $Na^+$, $NH_4^+$, $K^+$, $NO_3^-$ and $SO_4^{2-}$ as well as mean trajectory length, solar flux along the trajectory and mixing depth along the trajectory were used for PCA using IBM SPSS Statistics 21.0, and the results are presented in Table 3 and Table 4. If the compound concentration was below the detection limit, the data were replaced by a value half of the corresponding detection limit (Wold et al., 1987). Only factor loadings $|x| > 0.2$ were considered, and $|x| > 0.6$ were considered high loading and are depicted in bold.

(Page 15, Line 23-Page 16, Line 2)

*- A distinction is made between daytime and nighttime concentrations in the PCA analysis, and slightly different factors are identified. Please provide explanations for differences between all of the daytime and nighttime factors. (In the case of the nighttime factor 4, mixed marine and plastic burning emissions are suggested—please explain further and cite references.) Please consider agricultural activities as a possible emissions source.*

*(1) Please provide explanations for differences between all of the daytime and nighttime factors.*

Response: We have added explanations for differences between day and night factors as follows:

Day and night sources of DCRCs were similar, but there were some differences in source order and contribution. Anthropogenic activities followed by photochemical aging had a higher contribution during the day, which was probably related with higher BLHs during the day. Fuel combustion was the second most important source during the night, and its contribution was also higher during the night. Although secondary processing was the third most important source at night, its contribution was higher than that during the day, which may be related to more effective aqueous oxidation during the night. The daytime sources in PC4 and PC5 were not separated during the night.

(Page 17, Line 11-19)

*(2) In the case of the nighttime factor 4, mixed marine and plastic burning emissions are suggested—please explain further and cite references.*

Response: We have further explained night PC4 and cited references as follows:

The correlation between $C_9$ and $Na^+$ ($R^2 = 0.51$) suggests photooxidation of unsaturated fatty acids emitted from sea surface together with sea salt. As mentioned above, tPh is produced by wastes burning. High correlations of $C_9$, tPh, OC and $Na^+$ in PC4 may reveal a mixed aerosol source related to waste burning and photooxidation of unsaturated fatty acids emitted from the sea surface together with sea salt.

(Page 17, Line 5-10)

*(3) Please consider agricultural activities as a possible emissions source.*

Response: We have added agricultural activities as follows:

As shown in Table 3, PC1 was dominated by high loadings of $C_2$-$C_6$, $iC_5$, F, $hC_4$, Ph, $kC_3$, Pyr, $\omega C_2$, $\omega C_4$, Gly, MGly, OC and $K^+$, which were associated with anthropogenic activities (such as agricultural activities) followed by photochemical aging.

(Page 16, Line 4-7)

*11. The phrase "dicarboxylic acids and related compounds" is overused in the paper, and must be abbreviated for clarity. Please find an appropriate way to do so. An example might be "polar organic compounds (POCs)".*

Response: We have defined an abbreviation "DCRCs" for "dicarboxylic acids and related compounds", and have changed it throughout the draft.

*12. When reporting values summarizing the campaign data, be clear about whether the value is a mean, etc., in every case.*

Response: We have added "mean" or "total" when summary the campaign data. And have checked them throughout the manuscript.

*13. Throughout the document, please revise for grammar and accuracy of the wording. For example, on pg. 7, line 29, "trends" should be "concentrations".*

Response: We have changed "trends" to "concentrations", and revised grammar and accuracy of the wording throughout the manuscript.

*14. Please choose a consistent spelling and format for the following terms: "airmass", "daytime" vs. "day", "nighttime" vs. "night", "back trajectory" vs. "back-trajectory".*

Response: We have changed "air mass" to "airmass", "daytime" to "day", "nighttime" to "night", "back trajectory" to "back-trajectory" throughout the manuscript.

*15. Pg. 3, line 8: Is this really true that dicarboxylic acids and related compounds are typically studied in TSP rather than PM2.5? Please revisit.*

Response: We have deleted the old sentence and changed Pg. 3, line 8 in revised manuscript as follows:

Therefore, it is necessary to study the DCRC characteristics in $PM_{2.5}$.

(Page 3, Line 7-8)

*16. Section 3.2 (and throughout): The discussion of the contributors to "dicarboxylic acids and related compounds" would be much stronger with some context (rather than an empirical grouping of chemicals based on methods). What does this category of chemicals represent in the atmosphere? Could it be representative of water soluble organic carbon? Oxygenated organic species in general? Please support this with references.*

Response: DCRCs are not only oxygenated organic compounds but water-soluble components of SOA (Kawamura and Sakaguchi, 1999; Kawamura and Yasui, 2005; Pavuluri et al., 2010). Until now only a small fraction of water-soluble and oxygenated SOA can be identified in a compound specific manner. Zhu et al. (2017) reported that only 18.1-49.1% of SOA can be detected, a large fraction can't be identified. Therefore, DCRCs are important constituents in the water soluble and oxygenated matter of $PM_{2.5}$. We have changed related descriptions in the revised manuscript as follows:

Dicarboxylic acids and related compounds (oxocarboxylic acids and α-dicarbonyls) (DCRCs) are important constituents in $PM_{2.5}$, and mainly produced by secondary processes (Kawamura and Yasui, 2005; Pavuluri et al., 2010a). Due to their high water solubility, DCRCs contribute to the water soluble organic fraction of $PM_{2.5}$, which can have an impact on air quality (van Pinxteren et al., 2009; Kawamura and Bikkina, 2016; Kundu et al., 2010b).

(Page 3, Line 2-7)

PM$_{2.5}$ mass concentration at Mt. Tai during the campaign was 98.2 ±29.2, 98.6 ± 25.3 μg m$^{-3}$ in the day and night, respectively. DCRCs concentration contributed 1.2% and 1.1% to PM$_{2.5}$ in the day and night, respectively. The DCRCs-C at Mt. Tai in 2014 accounted for 3.24% and 3.20% of OC in the day and night, respectively. We have added these results in the revised manuscript as follows:

PM$_{2.5}$ mass concentration at Mt. Tai during the campaign was 98.2 ±29.2 and 98.6 ± 25.3 μg m$^{-3}$ during the day and night, respectively. DCRC total concentration contributed about 1.2% and 1.1% to PM$_{2.5}$ in the day and night, respectively. In addition, the DCRCs-C accounted for 3.3% and 3.2% of OC in the day and night, respectively.

(Page 9, Line 25-29)

Response: "Boundary layer height" has been abbreviated to "BLH" throughout the manuscript.

[revised manuscript text omitted]

---

## Editor Decision (ED1)

Editorial comments on ACP-2017-1240

Thank you for your efforts in responding to the reviewer's comments. The responses took care of some of the reviewer's original comments. However, there are a number of places where the responses were not adequate, as described below. The still requires significant revision before it will be acceptable for publication.

Comments on the author's response (page numbers from combined comment/response pdf)

Pages 2-3. The author's response to the comment the lack of day/night differences might be due to the fact that transport to the site takes several days is not really adequate. The authors need to admit the part of the reason for the lack of day/night difference is due to multi-day transport.

Page 3, Figure S1 doesn't add anything to the discussion, you could transmit the same information by noting the times of sunrise and sunset.

Page 3. What was different about the studies that did show day/night differences? Probably the nearby presence of the sources?

Page 5. When you say "pressure, temperature and RH didn't change much in clusters 2 and 4, are you saying between clusters 2 and 4, or in each cluster 2, and 4, over the timescale of the back trajectory?

Page 6. Yes, the reviewer was mistaken about the average RH.

Page 12. To use the ratio of PM2.5 to TSP to scale DCRCs and Levoglucosan, the way the authors have done requires the assumption that there is no size-dependent composition differences, so this needs to be stated as an assumption. Your response did not answer the reviewer's question about measurements from other locations.

Pages 14 and 15. The reviewers' comment that you have included too many significant figures in many places is correct, and your response is not correct. The appropriate number of significant figures should be based on the uncertainties of your measurements, which is a combination of the propagated errors, and detect limit. So, for example, your stated detection limits are between 0.05 and 0.1 ng/m3, and you do not specify what your propagated errors are, but let's assume they are ±10%, your uncertainty then would be ± the sum of 10% + the detection limit. So, in that case 3 significant figures are not justified. For numbers below 1 ng/m3, you can't justify more that 1 significant figure, since your detection limit is 0.1 ng/m3. Likewise, numbers in the 0.01 place are not significant, so those numbers in Tables 1 and 2 should be rounded to the nearest 0.1 place.

Page 15. The correct terms are "VOC sampling" and "VOC samples", using two plural terms (e.g. VOCs samples) is not correct, please those changes.

Pages 19 and 20. You did not describe the overall uncertainties in the measurements as requested. This needs to be done and then reflected in the reported data, i.e. significant figures.

Page 28. Point (5). Your answer does not really answer the question. The answer may be that transport to the site takes place over several day/night cycles.

Page 29 and 30. Removing a data point doesn't any more valid, and in fact could be interpreted as deceptive and misleading, and therefore, highly inappropriate.

Page 30, bottom. Do you have any iron measurements to back up your supposition about iron-oxalate photolysis?

Page 34. Your response to the reviewer's concern about significant figures is not acceptable. Your significant figures need to be based on measurement or estimate uncertainties. For example, 387 ng/m3 should be 390, due to both kinds of uncertainties.

Page 35. When you say "more than three times higher" Do you mean three times higher at Mt Tai?

Page 38 Point #3 at the bottom the reviewer wants to know what the values in the tables are. This should be in the title of the table. Also, you should note that the term "factor loading" means the correlation coefficient (r) between the variable (e.g. $Z_j$) and the principal component (PC#).

Page 39. The discussion of PCA results needs to be changed so that the PC numbers are connected to the names they have been given in the text. So Tables 3 and 4 the PC numbers have the names associated with them, and the numbers are given (e.g. PC2) after the name is given in the text.

Page 43. Aren't "coefficients of determination" and "factor loadings" the same thing?

Page 52, Figure 3. Is the boundary layer height above ground, or above sea level? If it is above ground, what site is the reference?

Comments on the corrected manuscript.

Page 7, Line 5. Please give the overall uncertainties here.

Page 8, Line 8. Here you should explain what "factor loading" is.

Page 10, Line 20. This assumes aerosol composition is not size-dependent, please note that and discuss how reasonable that assumption is.

Page 12, Lines 10-14. Mountain top sites are subject to "drainage flow" due to cooling of the ground surface and subsidence of the cooler air, that serves to pull air from above to the surface from above. Please consider what this might mean to your observations.

Page 17, line 2. Don't you mean that the "contribution of this source to the variance" was 13%?

---

## Author Response (AR2)

Dear Editor,

The authors would like to thank you for the good suggestions obtained in the review and for giving us the chance to further improve our manuscript! We have carefully considered all of your comments and revised the manuscript accordingly. Below, we provide the point-to-point response to your comments, with changes made in the manuscript highlighted in red.

Sincerely

Lingxiao Yang

Ph.D., Professor

Environment Research Institute

Shandong University

Jinan 250100

P. R. China

**Response to Editor comments on the author's response:**

*Thank you for your efforts in responding to the reviewer's comments. The responses took care of some of the reviewer's original comments. However, there are a number of places where the responses were not adequate, as described below. The still requires significant revision before it will be acceptable for publication.*

Response: We appreciate the editor for the comments and suggestions. We have revised the manuscript accordingly and here address the comments. For clarity, the editor's comments are listed below in black italics, while our responses and changes in the manuscript are shown in blue and red, respectively.

*1. Pages 2-3. The author's response to the comment the lack of day/night differences might be due to the fact that transport to the site takes several days is not really*

*adequate. The authors need to admit the part of the reason for the lack of day/night difference is due to multi-day transport.*

Response: We have admitted and added more discussion about multi-day transport being part of the reason for the lack of day/night difference in the revised manuscript as follows:

Furthermore, the predicted BLH (Fig. 3) suggests that the sampling site was mostly above the BLH during the sampling period, thus the impact of uplifted air on the 12h filter measurements should be minor. Moreover, it is noted that the summit of Mt. Tai is about a few hundred meters above other summits in the surrounding region (Fig. 4). Therefore, the airflow at Mt. Tai should be mainly influenced by the synoptic flow rather than drainage flows. Such an isolated mountain peak is often characterized by wind flows around the peak and small amounts of lifting over it. Nevertheless, under light wind conditions, sunlit mountain slopes may be a favored location for thermals lifting up air from lower levels. However, due to the predominant northwesterly winds, this might have only a minor effect on the performed measurements. No day-night variations of the DCRCs were observed, indicating similar air masses throughout the day and night measurement periods. Due to the fact that air masses arriving at Mt. Tai are transported over several days, multi-day transport has to be considered as part of the reason for the similar concentrations of the field samples taken during the day and night.

(Page 12, Line 27-Page 13, Line 12)

*2. Page 3, Figure S1 doesn't add anything to the discussion, you could transmit the same information by noting the times of sunrise and sunset.*

Response: We have deleted Fig. S1 and included the information about the time of sunrise and sunset in the revised manuscript as follows:

The time of sunrise and sunset in June at Mt. Tai was around 06:00 and 18:00, respectively. Therefore, 06:00-18:00 and 18:00-06:00 local time have been selected as the sampling times for day and night, respectively.

(Page 5, Line 11-14)

*3. Page 3. What was different about the studies that did show day/night differences?*

*Probably the nearby presence of the sources?*

Response: Pavuluri et al. (2010), Miyazaki et al. (2009) and Fu et al. (2008) have performed day and night sampling between 06:00-18:00 and 18:00-06:00 local time and showed different diurnal variations.

The reasons for the different diurnal variations in the three papers were as following:

Pavuluri et al. (2010) reported that due to the sea breeze effect, most of DCRCs presented much higher concentrations during the day. The sea breeze causes onshore flow of marine air masses in daytime, which are enriched with relatively fresh marine aerosols containing unsaturated fatty acids. They should produce diacids ($\geq C_4$) by photochemical oxidation.

Miyazaki et al. (2009) suggested that due to aqueous-phase oxidation and biomass burning, most of DCRCs had higher concentrations in the night samples.

Fu et al. (2008) reported that most of the organic compound classes showed higher concentrations in nighttime samples when organic aerosols can be transported over long distances, passing different source regions to the summit of Mt. Tai above the planetary boundary layer (PBL).

However, in this study and as noted above, the predicted BLH (Fig. 3) suggests that the sampling site was mostly above the BLH during the sampling period, thus the impact of uplifted air on the 12h filter measurements should be minor. Moreover, it is noted that the summit of Mt. Tai is about a few hundred meters above other summits in the surrounding region (Fig. 4). Therefore, the airflow at Mt. Tai should be mainly influenced by the synoptic flow rather than drainage flows. Such an isolated mountain peak is often characterized by wind flows around the peak and small amounts of lifting over it. Nevertheless, under light wind conditions, sunlit mountain slopes may be a favored location for thermals lifting up air from lower levels. However, due to the predominant northwesterly winds, this might have only a minor effect on the performed measurements. No day-night variations of the DCRCs were observed, indicating similar air masses throughout the day and night measurement periods. Due to the fact that air masses arriving at Mt. Tai are transported over several days,

multi-day transport has to be considered as important reason for the similar concentrations of the field samples taken during the day and night.

References:

Pavuluri, C.M., Kawamura, K., and Swaminathan, T. Water-soluble organic carbon, dicarboxylic acids, ketoacids, and a-dicarbonyls in the tropical Indian aerosols. Journal of Geophysical Research, 2010, 115.

Miyazaki, Y., Aggarwal, S.G., Singh, K., Gupta, P.K., Kawamura, K. Dicarboxylic acids and water-soluble organic carbon in aerosols in New Delhi, India, in winter: Characteristics and formation processes. Journal of Geophysical Research, 2009, 114.

Fu, P.Q., Kawamura, K., Okuzawa, K., Aggarwal, S.G., Wang, G.H., Kanaya, K., Wang, Z.F. Organic molecular compositions and temporal variations of summertime mountain aerosols over Mt. Tai, North China Plain, 2008. Journal of Geophysical Research, 2008, 113.

*4. Page 5. When you say "pressure, temperature and RH didn't change much in clusters 2 and 4, are you saying between clusters 2 and 4, or in each cluster 2, and 4, over the timescale of the back trajectory?*

Response: We have changed the sentence in the revised manuscript as follows:

We can see that the pressure, temperature and RH didn't change much in each of cluster 2 and cluster 4 over the timescale of the mean trajectories.
(Page 15, Line 6-7)

We also changed Fig. S1 caption as follows:

Fig. S1. Mean meteorological parameters along the mean trajectories in clusters 1 to 4.

*5. Page 6. Yes, the reviewer was mistaken about the average RH.*

Response: We corrected it as suggested, thanks a lot.

*6. Page 12. To use the ratio of PM2.5 to TSP to scale DCRCs and Levoglucosan, the way the authors have done requires the assumption that there is no size-dependent composition differences, so this needs to be stated as an assumption. Your response did not answer the reviewer's question about measurements from other locations.*

Response: We have added the statement of assumption as follows:

Deng et al. (2011) also showed that most of the water-soluble ions presented similar concentrations in $PM_{2.5}$ and TSP, and the ratios of their concentrations in $PM_{2.5}$ and TSP were more than 0.9. Therefore, we assumed there were small contributions of DCRCs from coarse mode particles.

(Page 10, Line 21-24)

We have added a reference from other locations in the revised manuscript as follows:

The low impact of particle size on particle composition has been reported at Mt. Gongga in China (Yang et al., 2009).

(Page 10, Line 24-26)

*7. Pages 14 and 15. The reviewers' comment that you have included too many significant figures in many places is correct, and your response is not correct. The appropriate number of significant figures should be based on the uncertainties of your measurements, which is a combination of the propagated errors, and detect limit. So, for example, your stated detection limits are between 0.05 and 0.1 ng/m3, and you do not specify what your propagated errors are, but let's assume they are ±10%, your uncertainty then would be ± the sum of 10% + the detection limit. So, in that case 3 significant figures are not justified. For numbers below 1 ng/m3, you can't justify more that 1 significant figure, since your detection limit is 0.1 ng/m3. Likewise, numbers in the 0.01 place are not significant, so those numbers in Tables 1 and 2 should be rounded to the nearest 0.1 place.*

Response: We have corrected the significant figures in Table 1 and 2, which were rounded to 0.1 place in the revised Table 1 and 2.

*8. Page 15. The correct terms are "VOC sampling" and "VOC samples", using two plural terms (e.g. VOCs samples) is not correct, please those changes.*

Response: We have corrected the two plural terms throughout the manuscript, and changed "VOCs samples" to "VOC samples".

*9. Pages 19 and 20. You did not describe the overall uncertainties in the measurements as requested. This needs to be done and then reflected in the reported*

*data, i.e. significant figures.*

Response: We have added description about overall uncertainties in the revised manuscript as follows:

Overall uncertainties for DCRC species were about 15% (see Boreddy et al., 2017 for details).

(Page 7, Line 7-8)

*10. Page 28. Point (5). Your answer does not really answer the question. The answer may be that transport to the site takes place over several day/night cycles.*

Response: We have admitted and added more discussion about multi-day transport being part of the reason for the lack of day/night difference in the revised manuscript as follows:

Furthermore, the predicted BLH (Fig. 3) suggests that the sampling site was mostly above the BLH during the sampling period, thus the impact of uplifted air on the 12h filter measurements should be minor. Moreover, it is noted that the summit of Mt. Tai is about a few hundred meters above other summits in the surrounding region (Fig. 4). Therefore, the airflow at Mt. Tai should be mainly influenced by the synoptic flow rather than drainage flows. Such an isolated mountain peak is often characterized by wind flows around the peak and small amounts of lifting over it. Nevertheless, under light wind conditions, sunlit mountain slopes may be a favored location for thermals lifting up air from lower levels. However, due to the predominant northwesterly winds, this might have only a minor effect on the performed measurements. No day-night variations of the DCRCs were observed, indicating similar air masses throughout the day and night measurement periods. Due to the fact that air masses arriving at Mt. Tai are transported over several days, multi-day transport has to be considered as part of the reason for the similar concentrations of the field samples taken during the day and night.

(Page 12, Line 27-Page 13, Line 12)

*11. Page 29 and 30. Removing a data point doesn't any more valid, and in fact could be interpreted as deceptive and misleading, and therefore, highly inappropriate.*

Response: The coeditor is right and, therefore, we have corrected this issue and have

added the point at ~1800 ng m$^{-3}$ C$_2$ and 35 μg m$^{-3}$ SO$_4^{2-}$ in Fig. 7. Moreover, we have changed the corresponding data in the revised manuscript.

The correlation coefficient was 0.28 for all daytime C$_2$ and SO$_4^{2-}$ concentrations. If we delete the point at ~1800 ng m$^{-3}$ C$_2$ and 35 μg m$^{-3}$ SO$_4^{2-}$, the correlation coefficient becomes 0.26. Therefore, the influence of this point was low.

*12. Page 30, bottom. Do you have any iron measurements to back up your supposition about ironoxalate photolysis?*

Response: We haven't performed iron measurements during the sampling period. So unfortunately, we can't back up the supposition about iron-oxalate photolysis. However, as a supplementary solution, we searched for measurements at Mt. Tai in the literature reporting TMI concentrations. Based on the literature, we have extended the related descriptions in the revised manuscript as follows:

Deng et al. (2011) and Shen et al. (2012) reported that Mt. Tai aerosol particles and cloud droplets include a substantial amount of transition metal ions, such as iron. Deng et al. (2011) reported that iron concentration was 0.71 μg m$^{-3}$ in PM$_{2.5}$ and 1.69 μg m$^{-3}$ in TSP during summer 2006. Moreover, Shen et al. (2012) reported that the average bulk cloud water concentration of iron was 44 μg L$^{-1}$ and 416 μg L$^{-1}$ during summer 2007 and 2008, respectively. Thus, iron-oxalate complex formation and photolysis might be possible chemical pathways occurring in Mt. Tai aerosols.

(Page 14, Line 11-18)

*13. Page 34. Your response to the reviewer's concern about significant figures is not acceptable. Your significant figures need to be based on measurement or estimate uncertainties. For example, 387 ng/m3 should be 390, due to both kinds of uncertainties.*

Response: We have corrected the significant figures in the revised manuscript as follows:

Using the ratio of PM$_{2.5}$/TSP (PM$_{2.5}$/TSP = 0.91) and DCRC concentrations in TSP at Mt. Tai in June 2006 (Kawamura et al., 2013), we have estimated the corresponding DCRC concentrations in PM$_{2.5}$ at Mt. Tai in June 2006 (1550, 220, 62 ng m$^{-3}$ for dicarboxylic acids, oxocarboxylic acids and α-dicarbonyls, respectively).

(Page 10, Line 26-Page 11, Line 1)

In addition, using the ratio of $PM_{2.5}$/TSP and the levoglucosan concentration in TSP at Mt. Tai in June 2006 (Fu et al., 2008), the estimated levoglucosan concentration in $PM_{2.5}$ at Mt. Tai in June 2006 was 390 ng m$^{-3}$. The result was more than five times higher than that in 2014 (levoglucosan: 70 ng m$^{-3}$) (Zhu et al., 2017), which suggests that biomass burning may have decreased from 2006 to 2014, or Mt. Tai was less influenced by emissions from lower altitudes during summer 2014.

(Page 11, Line 5-11)

*14. Page 35. When you say "more than three times higher" Do you mean three times higher at Mt Tai?*

Response: Thanks for the carefulness and your understanding is correct, we have corrected the expression as shown following:

The concentration of dicarboxylic acids at Mt. Tai in 2014 was similar to the concentration reported in 14 Chinese cities in 2003 (Ho et al., 2007), while oxocarboxylic acids and α-dicarbonyls were more than three times higher at Mt. Tai.

(Page 11, Line 14-17)

*15. Page 38 Point #3 at the bottom the reviewer wants to know what the values in the tables are. This should be in the title of the table. Also, you should note that the term "factor loading" means the correlation coefficient (r) between the variable (e.g. Zj) and the principal component (PC#).*

Response: According to the comment, we have changed the title of Table 3 and Table 4 as follows:

**Table 3.** PCA factor loadings for daytime DCRCs, OC, EC and inorganic ions as well as mean trajectory length, solar flux along trajectory and mixing depth along trajectory.

(Page 32)

**Table 4.** PCA factor loadings for nighttime DCRCs, OC, EC and inorganic ions as well as mean trajectory length, solar flux along trajectory and mixing depth along trajectory.

Furthermore, factor loading means the correlation coefficient between the variable and the principal component (PC). We have added descriptions about factor loading in the revised manuscript as follows:

Factor loading means the correlation coefficient between the variable and the PC, which reveals how much a variable contributes to the corresponding PC and how much a variable differs from others.

*16. Page 39. The discussion of PCA results needs to be changed so that the PC numbers are connected to the names they have been given in the text. So Tables 3 and 4 the PC numbers have the names associated with them, and the numbers are given (e.g. PC2) after the name is given in the text.*

Response: We have added the names of source types in Tables 3 and 4 as follows:

**Table 3.** PCA factor loadings for daytime DCRCs, OC, EC and inorganic ions as well as mean trajectory length, solar flux along trajectory and mixing depth along trajectory.

| Compounds | PC1[a] | PC2[b] | PC3[c] | PC4[d] | PC5[e] |
|---|---|---|---|---|---|
| $C_2$ | **0.854** | 0.382 | 0.203 | | |
| $C_3$ | **0.832** | 0.277 | | | |
| $C_4$ | **0.751** | 0.353 | 0.407 | | |
| $C_5$ | **0.764** | 0.267 | 0.437 | | |
| $C_6$ | **0.697** | 0.222 | 0.322 | | |
| $C_9$ | | -0.256 | 0.294 | **0.756** | 0.389 |
| $iC_5$ | **0.762** | | 0.523 | | |
| M | | | **0.885** | | |
| F | **0.630** | 0.288 | **0.635** | | |
| $hC_4$ | **0.794** | 0.205 | | | |
| Ph | **0.693** | | 0.431 | | 0.313 |
| tPh | | | | | **0.904** |

| Compounds | PC1[a] | PC2[b] | PC3[c] | PC4[d] | [e] |
|---|---|---|---|---|---|
| kC$_3$ | **0.716** | | 0.285 | | -0.202 |
| Pyr | **0.823** | 0.353 | 0.218 | | |
| ωC$_2$ | **0.854** | 0.406 | | | |
| ωC$_4$ | **0.881** | | | | |
| Gly | **0.834** | 0.396 | 0.248 | | |
| MGly | **0.687** | 0.540 | | | |
| OC | **0.787** | | | 0.559 | |
| EC | 0.411 | 0.226 | **0.632** | | -0.337 |
| Na$^+$ | 0.241 | 0.314 | | **0.862** | |
| NH$_4^+$ | 0.315 | **0.938** | | | |
| K$^+$ | **0.875** | 0.289 | | 0.293 | |
| NO$_3^-$ | 0.355 | **0.814** | 0.302 | | |
| SO$_4^{2-}$ | 0.279 | **0.895** | | | |
| Mean trajectory length | **-0.629** | **-0.627** | -0.255 | | |
| Solar flux along trajectory | -0.401 | 0.380 | | | |
| Mixing depth along trajectory | -0.507 | 0.393 | -0.302 | | |
| Variance (%) | 64% | 9% | 7% | 6% | 4% |

Extraction method: Principal Component Analysis (PCA).

Rotation method: varimax with Kaiser normalization.

[a] anthropogenic activities followed by photochemical aging

[b] secondary sources

[c] fuel combustion

[d] photooxidation of unsaturated fatty acids emitted from the sea surface together with sea salt

[e] waste burning

**Table 4.** PCA factor loadings for nighttime DCRCs, OC, EC and inorganic ions as well as mean trajectory length, solar flux along trajectory and mixing depth along trajectory.

| | | | | |
|---|---|---|---|---|
| $C_2$ | **0.674** | 0.504 | 0.464 | |
| $C_3$ | 0.341 | **0.728** | 0.436 | |
| $C_4$ | 0.356 | **0.678** | 0.506 | |
| $C_5$ | 0.578 | **0.699** | 0.285 | |
| $C_6$ | **0.661** | 0.400 | | 0.516 |
| $C_9$ | | 0.531 | | **0.726** |
| $iC_5$ | 0.407 | **0.657** | | 0.585 |
| M | | **0.870** | | 0.239 |
| F | 0.538 | **0.642** | 0.334 | |
| $hC_4$ | **0.735** | | 0.364 | |
| Ph | **0.610** | 0.478 | 0.305 | 0.467 |
| tPh | | | | **0.953** |
| $kC_3$ | 0.514 | **0.779** | | |
| Pyr | **0.834** | 0.293 | 0.356 | |
| $\omega C_2$ | **0.823** | 0.312 | 0.435 | |
| $\omega C_4$ | **0.893** | 0.261 | | 0.283 |
| Gly | **0.819** | 0.352 | 0.378 | |
| MGly | 0.568 | | **0.671** | |
| OC | **0.674** | 0.223 | | **0.660** |
| EC | | **0.770** | | |
| $Na^+$ | 0.374 | | | **0.865** |
| $NH_4^+$ | 0.273 | | **0.921** | |
| $K^+$ | **0.894** | 0.248 | | |
| $NO_3^-$ | 0.540 | -0.206 | **0.684** | |
| $SO_4^{2-}$ | | 0.365 | **0.887** | |
| Mean trajectory length | -0.564 | -0.408 | -0.531 | |
| Solar flux along trajectory | | | | |
| Mixing depth along trajectory | -0.522 | -0.427 | 0.293 | |
| Variance (%) | 56% | 14% | 13% | 6% |

Extraction method: Principal Component Analysis (PCA).

Rotation method: varimax with Kaiser normalization.

[a] anthropogenic activities followed by photochemical aging

[b] fuel combustion and photochemical reaction

[c] secondary processing

[d] a mixed aerosol source related to waste burning and photooxidation of unsaturated fatty acids emitted from the sea surface together with sea salt

*17. Page 43. Aren't "coefficients of determination" and "factor loadings" the same thing?*

Response: "coefficients of determination" is not the same as "factor loadings". Factor loading means the correlation coefficient between the variable and the PC, which reveals how much a variable contributes to the corresponding PC and how much a variable differs from others.

Coefficient of determination is $R^2$, for example from linear regression. We have reported coefficients of determination in the text except in the Figures as follows:

As shown in Fig. 5, DCRC concentrations exhibited weak and moderate correlations with total the concentration of selected DCRC precursors during the day ($R^2 = 0.29$) and night ($R^2 = 0.48$), respectively, where selected DCRC precursors included ethyne, ethene, isoprene, α-pinene, β-pinene, toluene, m/p-xylene and o-xylene (Warneck, 2003; Ervens et al., 2004; Bikkina et al., 2014; Tilgner and Herrmann, 2010).

(Page 13, Line 13-18)

As shown in Fig. 7, $C_2$ and $SO_4^{2-}$ exhibited a higher correlation during the night ($R^2 = 0.64$) than that during the day ($R^2 = 0.28$), and the linear regression slope during the night (0.028) was also higher than that during the day (0.016).

(Page 14, Line 3-6)

Dicarboxylic acids and $K^+$ exhibited a strong correlation during the first half of the measurement ($R^2 = 0.77$), while during the second half, dicarboxylic acids and $K^+$ exhibited no correlation ($R^2 = 0.04$) (Fig. 9).

(Page 15, Line 12-14)

*18. Page 52, Figure 3. Is the boundary layer height above ground, or above sea level? If it is above ground, what site is the reference?*

Response: In Fig. 3, the boundary layer height is above sea level, and the reference is sea level. We have added description in the revised manuscript as follows:

The reference height of the calculated BLH was sea level.

(Page 6, Line 17-18)

**Response to Editor comments on the corrected manuscript:**

*19. Page 7, Line 5. Please give the overall uncertainties here.*

Response: We have added a description about overall uncertainties in the revised manuscript as follows:

Overall uncertainties for DCRC species were about 15% (see Boreddy et al., 2017 for details).

(Page 7, Line 7-8)

*20. Page 8, Line 8. Here you should explain what "factor loading" is.*

Response: Factor loading means the correlation coefficient between the variable and the principal component (PC). We have explained "factor loading" in the revised manuscript as follows:

Factor loading means the correlation coefficient between the variable and the PC, which reveals how much a variable contributes to the corresponding PC and how much a variable differs from others.

(Page 8, Line 9-11)

*21. Page 10, Line 20. This assumes aerosol composition is not size-dependent, please note that and discuss how reasonable that assumption is.*

Response: We have added the statement of assumption as shown following:

Deng et al. (2011) also showed that most of the water-soluble ions presented similar concentrations in $PM_{2.5}$ and TSP, and the ratios of their concentrations in $PM_{2.5}$ and TSP were more than 0.9. Therefore, we assumed there were small contributions of DCRCs from coarse mode particles.

(Page 10, Line 21-24)

*22. Page 12, Lines 10-14. Mountain top sites are subject to "drainage flow" due to cooling of the ground surface and subsidence of the cooler air, that serves to pull air from above to the surface from above. Please consider what this might mean to your observations.*

Response: We have added a related discussion in the revised manuscript as follows:

During the day when the BLHs can be above the sampling site height, more polluted air can be transported from the lower (ground) levels to Mt. Tai top, while during the night, cooling of the ground surface and subsidence of cool air may pull down clean air masses from the free troposphere to the top of Mt. Tai (Fu et al., 2014). However, clear diurnal variations were not found in the DCRC concentrations. Furthermore, the predicted BLH (Fig. 3) suggests that the sampling site was mostly above the BLH during the sampling period, and thus the impact of uplifted air on the 12 h filter measurements should be minor. Moreover, it is noted that the summit of Mt. Tai is about a few hundred meters above other summits in the surrounding region (Fig. 4). Therefore, the airflow at Mt. Tai should be mainly influenced by the synoptic flow rather than drainage flows. Such an isolated mountain peak is often characterized by wind flows around the peak and small amounts of lifting over it. Nevertheless, under light wind conditions, sunlit mountain slopes may be a favored location for thermals lifting up air from lower levels. However, due to the predominant northwesterly winds, this might have only a minor effect on the performed measurements. No day-night variations of the DCRCs were observed, indicating similar air masses throughout the day and night measurement periods.

(Page 12, Line 22-Page 13, Line 12)

[Figure]

**Fig. 4.** Topographic map of Mt. Tai and the surrounding region. In the top and bottom panels the altitude is shown by the z-axis and by the color-map, respectively, both with units of meters. The digital SRTM (NASA's Shuttle Radar Topography Mission) elevation data are provided by the CIAT-CSI SRTM website (http://srtm.csi.cgiar.org).

*23. Page 17, line 2. Don't you mean that the "contribution of this source to the variance" was 13%?*

Response: We have changed the sentence in the revised manuscript as follows:

Moreover, the contribution of this source to the variance was higher during the night (13%) than that during the day (9%) suggesting that secondary processing was more important during the night.

(Page 18, Line 2-5)